# Fisher-Legendre (FishLeg) optimization of deep neural networks

**Jezabel R Garcia**[1*]**, Federica Freddi**[1*]**, Stathi Fotiadis**[1]**, Maolin Li**[1]**, Sattar Vakili**[1]**,
Alberto Bernacchia**[1*] **& Guillaume Hennequin**[1,2,*]

1. MediaTek Research, Cambourne Business Park, CB23 6DW, UK
   `first.last@mtkresearch.com`

2. Computational and Biological Learning Lab, Department of Engineering,
   University of Cambridge, Cambridge CB2 1PZ, UK
   `g.hennequin@eng.cam.ac.uk`

[*] Equal first/senior authors contributions

## Abstract

Incorporating second-order gradient information (curvature) into optimization can dramatically reduce the number of iterations required to train machine learning models. In natural gradient descent, such information comes from the Fisher information matrix which yields a number of desirable properties. As exact natural gradient updates are intractable for large models, successful methods such as KFAC and sequels approximate the Fisher in a structured form that can easily be inverted. However, this requires model/layer-specific tensor algebra and certain approximations that are often difficult to justify. Here, we use ideas from Legendre-Fenchel duality to learn a direct and efficiently evaluated model for the product of the inverse Fisher with any vector, in an online manner, leading to natural gradient steps that get more accurate over time despite noisy gradients. We prove that the resulting "Fisher-Legendre" (FishLeg) optimizer converges to a (global) minimum of non-convex functions satisfying the PL condition, which applies in particular to deep linear networks. On standard auto-encoder benchmarks, we show empirically that FishLeg outperforms standard first-order optimization methods, and performs on par with or better than other second-order methods, especially when using small batches. Thanks to its generality, we expect our approach to facilitate the handling of a variety neural network layers in future work.

## 1 Introduction & summary of contributions

The optimization of machine learning models often benefits from the use of second-order gradient (curvature) information, which can dramatically increase the per-iteration progress on the training loss. However, second-order optimizers for machine learning face a number of key challenges:

1. ML models tend to have many parameters, such that estimating the curvature along all dimensions is intractable. There are normally two ways around this: (i) using iterative methods where each iteration only exploits curvature information along a single dimension ("Hessian-free" methods; Martens et al., 2010), and (ii) developing custom curvature approximations that can be efficiently inverted to obtain parameter updates (Martens & Grosse, 2015; Grosse & Martens, 2016; Botev et al., 2017; George et al., 2018; Bahamou et al., 2022; Soori et al., 2022). The latter yields state-of-the-art performance in the optimization of deep networks (Goldfarb et al., 2020). However, development of structured curvature approximations must be done on a case-by-case basis for each architecture (e.g. fully-connected or convolutional layer) and requires mathematical assumptions that are difficult to justify.

2. ML datasets tend to be large as well, such that the loss, its gradients, and its curvature can be only stochastically estimated from mini-batches. While noise can be mitigated by a combination of large mini-batches and momentum, estimating the various components of the curvature matrix before inverting it (as opposed to estimating the inverse directly) introduces a bias that can potentially be detrimental.

Here, we focus primarily on the Fisher information matrix (FIM) as a notoriously effective source of curvature information for training ML models. For probabilistic models, preconditioning the gradient by the inverse FIM yields so-called natural gradient updates (Amari, 1998). We introduce a novel framework for second-order optimization of high-dimensional probabilistic models in the presence of gradient noise. Instead of approximating the FIM in a way that can easily be inverted (e.g. as in KFAC; Martens & Grosse, 2015 and related approaches), we directly parameterize the inverse FIM. We make the following contributions:

- We show that the inverse FIM can be computed through the Legendre-Fenchel conjugate of a cross entropy between model distributions.
- We provide an algorithm (FishLeg) which meta-learns the inverse FIM in an online fashion, and we prove convergence of the corresponding parameter updates.
- We first study its application to deep linear networks, an example of non-convex and pathologically curved loss functions with a Fisher matrix known in closed form (Bernacchia et al., 2018, Huh, 2020), and find that convergence occurs much faster than SGD with momentum or Adam.
- We then show that, in standard auto-encoders benchmarks, FishLeg operating on a block-diagonal Kronecker parameterization of the inverse Fisher performs similarly to – and sometimes outperforms – previous approximate natural gradient methods (Goldfarb et al., 2020), whilst being only twice slower than SGD with momentum in wall-clock time per iteration.

Similar to Amortized Proximal Optimization (Bae et al., 2022), FishLeg can accommodate arbitrary parameterizations of the inverse Fisher, thus facilitating future applications of the natural gradient to a broad range of network architectures where manually approximating the FIM in an easily invertible form is otherwise difficult.

## 2 BACKGROUND

### 2.1 FISHER INFORMATION AND THE NATURAL GRADIENT

We consider a probabilistic model parameterized by a vector $\boldsymbol{\theta}$, which attributes a negative log-likelihood $\ell(\boldsymbol{\theta}, \mathcal{D}) = -\log p(\mathcal{D}|\boldsymbol{\theta})$ to any collection $\mathcal{D}$ of data points drawn from a data distribution $p^\star(\mathcal{D})$. This covers a broad range of models, including discriminative models for regression or classification, as well as generative models for density modelling. The goal is to find parameters $\boldsymbol{\theta}^\star$ to approximate the true data distribution $p^\star(\mathcal{D})$ by the model distribution $p(\mathcal{D}|\boldsymbol{\theta}^\star)$.

The Fisher information matrix (FIM) measures how much information can be obtained about parameters $\boldsymbol{\theta}$ after observing data $\mathcal{D}$ under the model $p(\mathcal{D}|\boldsymbol{\theta})$, and captures redundancies between parameters (Rao, 1992). The FIM is defined as

$$\mathcal{I}(\boldsymbol{\theta}) = \mathbb{E}_{\mathcal{D}\sim p(\mathcal{D}|\boldsymbol{\theta})}\nabla_{\boldsymbol{\theta}}\ell(\boldsymbol{\theta}, \mathcal{D})\nabla_{\boldsymbol{\theta}}\ell(\boldsymbol{\theta}, \mathcal{D})^\top. \tag{1}$$

By this definition, the FIM is a positive semi-definite matrix. It can be shown that under certain regularity conditions, and if $\ell$ is twice differentiable w.r.t. $\boldsymbol{\theta}$, then the FIM can also be computed as

$$\mathcal{I}(\boldsymbol{\theta}) = \mathbb{E}_{\mathcal{D}\sim p(\mathcal{D}|\boldsymbol{\theta})}\nabla_{\boldsymbol{\theta}}^2\ell(\boldsymbol{\theta}, \mathcal{D}) \tag{2}$$

It is important to note that the average is computed over the model distribution $p(\mathcal{D}|\boldsymbol{\theta})$, *not* the data distribution $p^\star(\mathcal{D})$. Averaging Eq. 1 over $p^\star(\mathcal{D})$ results in the *empirical Fisher* matrix (Kunstner et al., 2019), while averaging Eq. 2 over $p^\star(\mathcal{D})$ results in the Hessian of the loss. The FIM, the empirical Fisher and the Hessian are all different and sometimes confused (Thomas et al., 2020).

We consider cases in which parameters $\boldsymbol{\theta}$ are obtained by maximum likelihood:

$$\boldsymbol{\theta}^\star = \arg\min_{\boldsymbol{\theta}} \mathcal{L}(\boldsymbol{\theta}) \tag{3}$$

where the population loss function is defined as

$$\mathcal{L}(\boldsymbol{\theta}) = \mathbb{E}_{\mathcal{D}\sim p^\star}\ell(\boldsymbol{\theta}, \mathcal{D}) \tag{4}$$

which is in general non-convex, in particular when the model distribution is parameterised by deep neural networks. The natural gradient update takes the form

$$\boldsymbol{\theta}_{t+1} = \boldsymbol{\theta}_t - \eta\,\mathcal{I}(\boldsymbol{\theta})^{-1}\mathbf{g}(\boldsymbol{\theta}) \quad \text{with} \quad \mathbf{g}(\boldsymbol{\theta}) = \nabla_{\boldsymbol{\theta}}\mathcal{L}(\boldsymbol{\theta}) \tag{5}$$

where $\eta$ is a learning rate and $\mathbf{g}$ is the gradient of the loss. In practice, the true distribution of data $p^\star(\mathcal{D})$ is unknown and we have only access to the empirical loss and a stochastic estimate $\hat{\mathbf{g}}$ of the gradient based on a finite sample of data $\mathcal{D} \sim p^\star$. Similarly, while the FIM may be sometimes computed exactly (Bernacchia et al., 2018, Huh, 2020), in most circumstances it is also estimated from a mini-batch $\mathcal{D} \sim p$. To guarantee invertibility of the FIM, a small amount of "damping" can be added to it when it is nearly singular. Note that Eq. 5 is identical to gradient descent when the FIM equals the identity matrix. The natural gradient has several nice properties that make it an efficient optimizer (Amari, 1998), but computing and inverting the FIM is usually costly.

## 2.2 THE LEGENDRE-FENCHEL CONJUGATE

The Legendre-Fenchel (LF) conjugate is a useful tool in optimization theory to map pairs of problems via duality (Boyd & Vandenberghe, 2004) and was introduced in the context of mechanics and thermodynamics (Zia et al., 2009). In the following, we deviate from the standard textbook notation to avoid conflicts with the rest of the paper and to facilitate the translation of these classical results to our problem. Consider a twice differentiable function $\mathcal{H}(\boldsymbol{\delta})$ of a vector $\boldsymbol{\delta}$, generally non-convex and assume a minimum of $\mathcal{H}$ exists. The LF conjugate is equal to

$$\mathcal{H}^\star(\mathbf{u}) = \min_{\boldsymbol{\delta}} \mathcal{H}(\boldsymbol{\delta}) - \mathbf{u}^T \boldsymbol{\delta}. \tag{6}$$

The LF conjugate, also known as the convex conjugate, is defined also for non-differentiable functions and is always convex. We summarize here two properties of the convex conjugate $\mathcal{H}^\star$ that we use in our derivations (see chapter 3.3 of Boyd & Vandenberghe, 2004 and Zia et al., 2009 for details):

**Property 1** The gradient of the conjugate is equal to

$$\nabla_{\mathbf{u}} \mathcal{H}^\star(\mathbf{u}) = \tilde{\boldsymbol{\delta}}(\mathbf{u}) \qquad \text{where} \quad \tilde{\boldsymbol{\delta}}(\mathbf{u}) = \operatorname{argmin}_{\boldsymbol{\delta}} \mathcal{H}(\boldsymbol{\delta}) - \mathbf{u}^T \boldsymbol{\delta}. \tag{7}$$

The minimizer $\tilde{\boldsymbol{\delta}}(\mathbf{u})$ also satisfies $\nabla_{\boldsymbol{\delta}} \mathcal{H}(\tilde{\boldsymbol{\delta}}(\mathbf{u})) = \mathbf{u}$, implying that $\nabla_{\mathbf{u}} \mathcal{H}^\star$ is a (local) inverse function of $\nabla_{\boldsymbol{\delta}} \mathcal{H}$. The inverse is global when the function $\mathcal{H}$ is strictly convex:

$$\nabla_{\mathbf{u}} \mathcal{H}^\star(\mathbf{u}) = (\nabla_{\boldsymbol{\delta}} \mathcal{H})^{-1}(\mathbf{u}). \tag{8}$$

**Property 2** The Hessian matrix of the conjugate is equal to the inverse (if it exists) of the Hessian matrix of $\mathcal{H}$, computed at $\tilde{\boldsymbol{\delta}}(\mathbf{u})$:

$$\nabla_{\mathbf{u}}^2 \mathcal{H}^\star(\mathbf{u}) = \left(\nabla_{\boldsymbol{\delta}}^2 \mathcal{H}(\tilde{\boldsymbol{\delta}}(\mathbf{u}))\right)^{-1}. \tag{9}$$

## 3 RELATED WORK

**Natural gradient** The Hessian of non-convex losses is not positive definite, and second-order methods such as Newton's update typically do not converge to a minimum (Dauphin et al., 2014). The natural gradient method substitutes the Hessian with the FIM, which is positive (semi)definite (Amari, 1998). The natural gradient has been applied to deep learning in recent years (Martens & Grosse, 2015, Bernacchia et al., 2018, Huh, 2020, Kerekes et al., 2021), and has been proved approximately equivalent to local loss optimization (Benzing, 2022, Meulemans et al., 2020, Amid et al., 2022). Previous work on natural gradient approximately estimated and inverted the FIM analytically (Martens & Grosse, 2015; Martens et al., 2018). Instead, we use an exact formulation of the inverse FIM as the Hessian of the LF conjugate of the cross entropy, and we provide an algorithm to meta-learn an approximation to the inverse FIM during training.

**Legendre-Fenchel conjugate** For convex losses, Eq. 8 implies that the parameter update $\Delta\boldsymbol{\theta} = -\nabla_{\mathbf{g}} \mathcal{L}^\star(\mathbf{g}(\boldsymbol{\theta})) + \nabla_{\mathbf{g}} \mathcal{L}^\star(\mathbf{0})$ converges to a minimum in one step (Maddison et al., 2021; Chraibi et al., 2019). However, this update is not practical because finding $\mathcal{L}^\star$ is at least as hard as finding a minimum of the loss. Chraibi et al. (2019) propose to learn $\mathcal{L}^\star$ by a neural network, and similar "amortized duality" approaches have recently been developed for optimal transport (Nhan Dam et al., 2019; Korotin et al., 2019; Makkuva et al., 2020; Amos, 2022). Maddison et al. (2021) prove that, if a surrogate for $\mathcal{L}^\star$ satisfies relative smoothness in dual space, then linear convergence rates can be proven for non-Lipschitz or non-strongly convex losses. Here we consider non-convex losses and we use the LF conjugate of the cross entropy for meta-learning the natural gradient update.

**Meta-learning optimizers** Many standard optimizers have comparable performance when appropriately tuning a handful of static hyperparameters (Schmidt et al., 2021). *Learned* optimizers often outperform well-tuned standard ones by tuning a larger number of hyperparameters online during training (Chen et al., 2021; Hospedales et al., 2021; Bae et al., 2022). Different methods for learning optimizers have been proposed, including hypergradients (Maclaurin et al., 2015; Andrychowicz et al., 2016; Franceschi et al., 2017; Wichrowska et al., 2017; Wu et al., 2018; Park & Oliva, 2019; Micaelli & Storkey, 2021), reinforcement learning (Li & Malik, 2016; Bello et al., 2017), evolution strategies (Metz et al., 2019; Vicol et al., 2021), and implicit gradients (Lorraine et al., 2020; Rajeswaran et al., 2019; Clarke et al., 2021). In our work, hyperparameters do not minimize the loss, instead they minimize an auxiliary loss designed to learn the natural gradient update.

# 4 FISHER-LEGENDRE OPTIMIZATION

## 4.1 COMPUTATION OF THE NATURAL GRADIENT VIA THE LEGENDRE-FENCHEL CONJUGATE

We use the properties of the LF conjugate, in particular Eq. 9, to directly learn the *inverse* of the (damped) FIM, and use it in the natural gradient update of Eq. 5. This is different from previous work on natural gradient, which aimed at approximating the FIM analytically and then inverting it. We prove the following:

**Theorem 1.** *Assume that the negative log-likelihood $\ell(\boldsymbol{\theta}, \mathcal{D}) = -\log p(\mathcal{D}|\boldsymbol{\theta})$ is twice differentiable. Let $\gamma > 0$ be a small damping parameter. Define the regularized cross entropy between $p(\mathcal{D}|\boldsymbol{\theta})$ and $p(\mathcal{D}|\boldsymbol{\theta} + \boldsymbol{\delta})$*

$$\mathcal{H}_\gamma(\boldsymbol{\theta}, \boldsymbol{\delta}) = \mathbb{E}_{\mathcal{D} \sim p(\mathcal{D}|\boldsymbol{\theta})} \ell(\boldsymbol{\theta} + \boldsymbol{\delta}, \mathcal{D}) + \frac{\gamma}{2} \|\boldsymbol{\delta}\|^2 \tag{10}$$

*and the function*

$$\tilde{\boldsymbol{\delta}}_\gamma(\boldsymbol{\theta}, \mathbf{u}) = \mathrm{argmin}_{\boldsymbol{\delta}} \ \mathcal{H}_\gamma(\boldsymbol{\theta}, \boldsymbol{\delta}) - \mathbf{u}^T \boldsymbol{\delta}. \tag{11}$$

*Then, the inverse damped Fisher information matrix exists and is equal to*

$$(\mathcal{I}(\boldsymbol{\theta}) + \gamma I)^{-1} = \nabla_{\mathbf{u}} \tilde{\boldsymbol{\delta}}_\gamma(\boldsymbol{\theta}, \mathbf{0}). \tag{12}$$

*Proof.* Here we provide a sketch of the proof, see Appendix A.2 for the full proof. Lemma 1 proves that the regularized cross-entropy in Eq. 10 has a unique minimum at $\boldsymbol{\delta} = \mathbf{0}$, therefore $\tilde{\boldsymbol{\delta}}_\gamma(\boldsymbol{\theta}, \mathbf{0}) = \mathbf{0}$. The proof continues by expressing the damped FIM as the Hessian of the regularized cross-entropy, which is done by reparameterizing Eq. 2. Next, we use Eq. 9 to express the inverse damped FIM as the Hessian of the LF conjugate of the regularized cross-entropy (seen as a function of $\boldsymbol{\delta}$). From there, the theorem follows from Eq. 7. □

To clarify the connection with the LF conjugate, we note that Eq. 11 represents the gradient of the LF conjugate of the regularized cross-entropy, while Eq. 12 is the Hessian of the LF conjugate – see Appendix A.2 for details, and Appendix A.3 for an informal mathematical argument that helps understanding Theorem 1.

Our goal is to compute the damped natural gradient update by solving the optimization problem in Eq. 11, and substituting the inverse damped FIM computed by equation 12 into the definition of the natural gradient step in Eq. 5, yielding

$$\boldsymbol{\theta}_{t+1} = \boldsymbol{\theta}_t - \eta \, \nabla_{\mathbf{u}} \tilde{\boldsymbol{\delta}}_\gamma(\boldsymbol{\theta}_t, \mathbf{0}) \mathbf{g}(\boldsymbol{\theta}_t). \tag{13}$$

Importantly, note that Eq. 11 does not need to be solved for all possible pairs of $\boldsymbol{\theta}$ and $\mathbf{u}$. Indeed, given the current value of parameters $\boldsymbol{\theta}_t$, it is sufficient to compute $\tilde{\boldsymbol{\delta}}_\gamma(\boldsymbol{\theta}_t, \mathbf{u})$ (i) at small values of $\mathbf{u}$, because $\nabla_{\mathbf{u}} \tilde{\boldsymbol{\delta}}_\gamma$ is evaluated at $\mathbf{u} = \mathbf{0}$ in Eq. 12, and (ii) along the direction $\mathbf{u} \propto \mathbf{g}(\boldsymbol{\theta}_t)$, because $\nabla_{\mathbf{u}} \tilde{\boldsymbol{\delta}}_\gamma$ is multiplied by $\mathbf{g}(\boldsymbol{\theta}_t)$ in Eq. 13 such that knowing the slope of $\tilde{\boldsymbol{\delta}}_\gamma(\boldsymbol{\theta}_t, \mathbf{u})$ along any other direction would be irrelevant. Nevertheless, computing Eq. 11 for such restricted $(\boldsymbol{\theta}, \mathbf{u})$ pairs may still not be easier than computing the FIM by standard methods, e.g. by computing Eq. 1 and inverting the resulting matrix. In section 4.2, we therefore propose a practical instantiation of the ideas developed in this section, which yields an online algorithm with provable convergence guarantees.

### 4.2 ONLINE META-LEARNING OF NATURAL GRADIENT UPDATES

In principle, Theorem 1 provides an exact implementation of natural gradient by solving the optimization problem in Eq. 11 and then applying the update of Eq. 13. However, this approach is impractical due to the complexity of solving Eq. 11. To make it practical, we propose learning an approximation $\overline{\delta}_\gamma(\theta, \mathbf{u}, \lambda)$ of the true $\tilde{\delta}_\gamma(\theta, \mathbf{u})$, through a set of *auxiliary* parameters $\lambda$. The auxiliary parameters $\lambda$ are updated *online*, during optimization, to push $\overline{\delta}_\gamma$ towards a solution of Eq. 11 but without requiring convergence. This is akin to a meta-learning approach whereby only one or a few steps are taken towards a desired solution for meta-parameters $\lambda$, which are optimized concurrently with parameters $\theta$ in an inner loop (Hospedales et al., 2021). Here we choose gradient-based meta-learning updates (Finn et al., 2017), but other choices are possible, e.g. implicit differentiation (Lorraine et al., 2020, Rajeswaran et al., 2019) or evolutionary strategies (Metz et al., 2019,Gao & Sener, 2022).

For the auxiliary parameterization, we choose a linear function of $\mathbf{u}$,

$$\overline{\delta}_\gamma(\theta, \mathbf{u}, \lambda) = Q(\lambda)\mathbf{u}, \tag{14}$$

where $Q(\lambda)$ is a positive definite matrix which will thus effectively estimate the inverse damped FIM (see equation 12). Appropriate choices for $Q$ should take into account the architecture of the model, as discussed further below. Although the r.h.s. of Eq. 14 does not depend on $\theta$ explicitly, an implicit dependence will arise from us learning the parameters $\lambda$ in a way that depends on the momentary model parameters $\theta$. Indeed, for learning $\lambda$, we perform gradient descent using Adam on the following auxiliary loss $\mathcal{A}$, a choice justified by Eq. 11:

$$\mathcal{A}_\gamma(\theta, \mathbf{u}, \lambda) = \mathcal{H}_\gamma\left(\theta, Q(\lambda)\mathbf{u}\right) - \mathbf{u}^T Q(\lambda)\mathbf{u}. \tag{15}$$

In summary, we alternate between updating $\lambda$ to minimize the auxiliary loss in Eq. 15, and updating $\theta$ according to Eq. 13 to minimize the main loss,

$$\lambda_{t+1} = \lambda_t - \alpha \, \text{AdamUpdate}\left(\nabla_\lambda \mathcal{A}_\gamma(\theta_t, \epsilon \mathbf{g}(\theta_t), \lambda_t)\right), \tag{16}$$

$$\theta_{t+1} = \theta_t - \eta \, Q(\lambda_{t+1}) \, \mathbf{g}(\theta_t), \tag{17}$$

where $\alpha$ is a learning rate for the auxiliary parameters and $\epsilon$ is a small scalar (both are hyperparameters). Note that the auxiliary loss is computed at $\epsilon \mathbf{g}(\theta)$ because, as argued in section 4.1, it is sufficient to compute $\overline{\delta}$ at small values of $\mathbf{u}$ in the direction of $\mathbf{u} \propto \mathbf{g}(\theta_t)$. In Appendix A.9, we provide an analysis of these coupled dynamics in a simple linear-Gaussian model.

We parameterize $Q(\lambda)$ as a positive definite Kronecker-factored block-diagonal matrix, with block sizes reflecting the layer structure of deep fully-connected neural networks (section A.5). Although this is similar to parameterizations used in previous studies of natural gradient (Martens & Grosse, 2015), we stress that our approach is more flexible since it learns the inverse FIM rather than approximating and inverting the FIM analytically. In fact, we show in the appendix A.6 that alternative forms of the matrix $Q$, different from those used before, may provide a better approximation of the inverse FIM (see Fig. 4).

### 4.3 CONVERGENCE OF ONLINE NATURAL GRADIENT

In this section we prove that the update of Eq. 17 converges to a minimum of the loss for the class of PL functions (Polyak-Lojasiewicz), which are generally non-convex. Charles & Papailiopoulos (2018) proved that deep linear networks, which are non-convex (Saxe et al., 2013), are PL almost everywhere. We study deep linear networks empirically in section 5.1. We stress that the goal of this section is to prove convergence, rather than to obtain tight bounds on the convergence rate.

We provide convergence guarantees for both the true gradient $\mathbf{g}$ and stochastic gradients $\hat{\mathbf{g}}$. Theorem 2 is a special case of Theorem 1 in Radhakrishnan et al. (2020), while Theorem 3 is new to our knowledge. In both cases, the crucial assumption is that $Q$ is a positive definite matrix, which holds for our chosen parameterization (see section A.5). For details on definitions and proofs, see appendix A.4.

**Theorem 2.** *Assume the loss function $\mathcal{L}$ is $\xi$-smooth and $\mu$-PL. Let $\theta^* \in \arg\min_\theta \mathcal{L}(\theta)$. Assume that the eigenvalues of $Q(\lambda_t)$ are lower- and upper-bounded uniformly in time by $\lambda_{min} > 0$ and*

$\lambda_{max}$, respectively. We have the following rate of convergence for optimization using the update rule of Eq. 17 with $\eta = \frac{1}{\xi\lambda_{\max}}$:

$$\mathcal{L}(\boldsymbol{\theta}_t) - \mathcal{L}(\boldsymbol{\theta}^*) \leq \left(1 - \frac{\mu\lambda_{\min}}{2\xi\lambda_{\max}}\right)^t (\mathcal{L}(\boldsymbol{\theta}_0) - \mathcal{L}(\boldsymbol{\theta}^*)). \tag{18}$$

**Theorem 3.** *Let the same assumptions as in Theorem 2 hold. In addition, assume that* $\mathbb{E}[\|\hat{\mathbf{g}}(\boldsymbol{\theta})\|^2] \leq G^2$. *We have the following rate for optimization using update rule 17 with* $\eta_t = \frac{2}{\mu\lambda_{\min}(t+1)}$:

$$\mathcal{L}(\boldsymbol{\theta}_t) - \mathcal{L}(\boldsymbol{\theta}^*) \leq \frac{A}{t+1} \quad where \quad A = \max\left\{\frac{2\xi\lambda_{\max}^2 G^2}{\mu^2\lambda_{\min}^2}, \mathcal{L}(\boldsymbol{\theta}_0) - \mathcal{L}(\boldsymbol{\theta}^*)\right\}. \tag{19}$$

## 4.4 DETAILS OF THE ALGORITHM AND FURTHER REMARKS

Here we detail our practical implementation of the online FishLeg algorithm presented in section 4.2, also summarized in Algorithm 1. Our code is available here on GitHub.

**Auxiliary optimization** Since $\mathcal{H}_\gamma$ is evaluated on a model-sampled mini-batch, it is a noisy quantity whose variance depends on the hyperparameter $\epsilon$. While this variance could be reduced e.g. by using an antithetic estimator (Gao & Sener, 2022), we opted instead for analytically taking the small $\epsilon$ limit of the auxiliary loss. By Taylor expanding equation 15 (use equations 23, 24), and dropping terms that are constant in $\boldsymbol{\lambda}$, we arrive at

$$\tilde{\mathcal{A}}_\gamma(\boldsymbol{\theta}, \mathbf{u}, \boldsymbol{\lambda}) \equiv \mathbf{u}^T \left[\frac{1}{2}Q(\boldsymbol{\lambda})\nabla_{\boldsymbol{\delta}}^2\mathcal{H}_\gamma(\boldsymbol{\theta}, \mathbf{0})Q(\boldsymbol{\lambda}) - Q(\boldsymbol{\lambda})\right]\mathbf{u}. \tag{20}$$

The auxiliary update of Eq. 16 thus becomes $\boldsymbol{\lambda}_{t+1} = \boldsymbol{\lambda}_t - \alpha\,\text{AdamUpdate}(\nabla_{\boldsymbol{\lambda}}\tilde{\mathcal{A}}_\gamma(\boldsymbol{\theta}_t, \mathbf{g}/\|\mathbf{g}\|, \boldsymbol{\lambda}_t))$ where the normalization of $\mathbf{g}$ is important for continuing to learn about curvature even when gradients are small. In order to avoid differentiating through the Hessian-vector product in Eq. 20, we note that the gradient of $\tilde{\mathcal{A}}$ w.r.t. $\boldsymbol{\lambda}$ can be written as

$$\nabla_{\boldsymbol{\lambda}}\tilde{\mathcal{A}}(\boldsymbol{\theta}, \mathbf{u}, \boldsymbol{\lambda}) = \left(\nabla_{\boldsymbol{\delta}}^2\mathcal{H}(\boldsymbol{\theta}, \mathbf{0})Q(\boldsymbol{\lambda})\mathbf{u} - \mathbf{u}\right)^\top \nabla_{\boldsymbol{\lambda}}\left[Q(\boldsymbol{\lambda})\mathbf{u}\right]. \tag{21}$$

Algorithm 1 in the appendix shows how to implement this efficiently using automatic differentiation tools. As a side note, Eq. 21 makes it clear that $Q$ will get to approximate the inverse of $\mathcal{I}(\boldsymbol{\theta}) + \gamma I$ (which is also the Hessian of the regularized cross-entropy, see equation 24) at least in the relevant subspace occupied by $\mathbf{u} = \mathbf{g}/\|\mathbf{g}\|$ over iterations. Moreover, note that the noise in $\mathcal{H}_\gamma$ simply adds stochasticity to the auxiliary loss but does not bias its gradient w.r.t. $\boldsymbol{\lambda}$. Finally, Eq. 20 reveals a connection with Hessian-free optimization which we discuss in Appendix A.8.

**Damping** Practitioners of second-order optimization in deep learning have consistently found it critical to apply damping when the FIM becomes near-singular. Although adaptive damping schemes exist and could be incorporated in FishLeg, here we followed Goldfarb et al. (2020) and used static damping; specifically, we used a fixed $\gamma$ in Eq. 10, treated as a tuned hyperparameter.

**Momentum** We have found useful to use momentum as many other optimizers do (e.g. KFAC, KBFGS). In most of our experiments, we implemented momentum on $\mathbf{g}$ (as per line 16 of Algorithm 1). For our wall-clock time results (Fig. 3), however, we applied momentum to our natural gradient estimate $Q(\boldsymbol{\lambda})\mathbf{g}$ instead, in order to preserve the low-rank structure of $\mathbf{g}$ at each layer of the neural network. This enabled us to substantially speed up the computation of $Q(\boldsymbol{\lambda})\mathbf{g}$ on small mini-batches, but did not affect training and test errors noticeably.

**Initialization of $\boldsymbol{\lambda}$** We find it useful to initialize $\boldsymbol{\lambda}$ such that $Q(\boldsymbol{\lambda}_0) = \frac{\eta_{\text{SGDm}}}{\eta} \times I$, where $\eta_{\text{SGDm}}$ is a learning rate that is known to give good results with SGD-with-momentum. With such an initialization, FishLeg therefore initially behaves like SGDm. Future work could investigate smarter initialization / warm-starting schemes, e.g. starting $Q(\boldsymbol{\lambda})$ in an Adam-like diagonal approximation of the empirical Fisher.

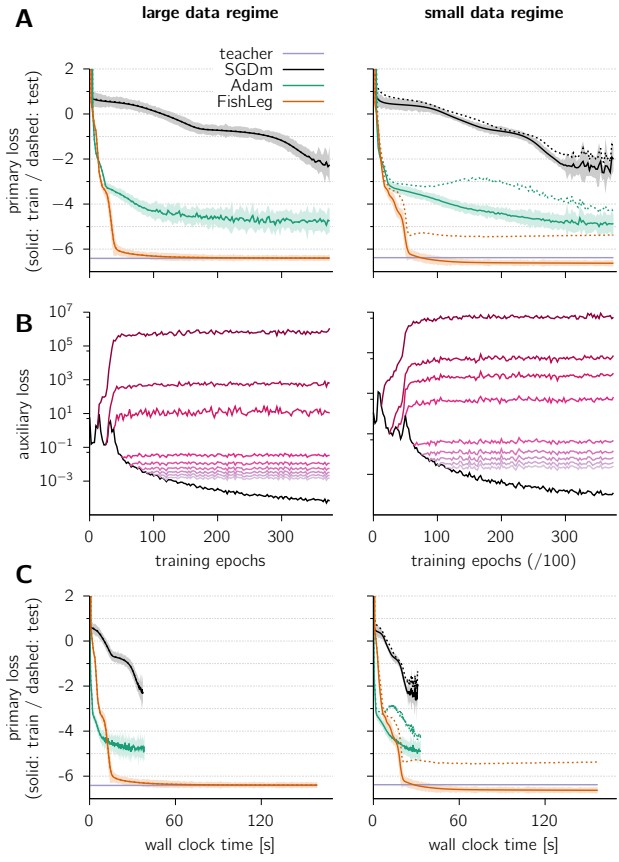

Figure 1: **Application to deep linear networks.** A deep linear network (20 layers of size 20 each) with Gaussian output likelihood is trained on a number of data samples (left: 4000; right: 40) generated by a noisy teacher in the same model class. **(A)** Loss evaluated on the training (solid) and testing set (dashed), as a function of training epoch. The gray line shows the noise floor, i.e. the test loss evaluated on the teacher network. Lines show mean across 10 independent experiments (teacher, initial conditions, etc), shadings show $\pm 1$ stdev. **(B)** Evolution of the auxiliary loss during training (black), with Fisher-matrix-vector products evaluated exactly according to the analytical expression given in (Bernacchia et al., 2018). To assess how well the auxiliary optimization steps minimize the auxiliary loss – a moving target when $\theta$ changes –, the auxiliary parameters $\lambda$ are frozen every 500 iterations and subsequently used to re-evaluate the auxiliary loss (colored lines). **(C)** Same as (A), here as a function of wall-clock time. Parameters: minibatch size = 40, $\eta = 0.04$, $\alpha = 0.001$, $\beta = 0.9$, $\eta_{\text{SGDm}} = 0.002$, $\eta_{\text{Adam}} = 0.0002$.

## 5 EMPIRICAL EVALUATION

### 5.1 DEEP LINEAR NETWORKS

We first studied FishLeg in the context of deep linear networks, whose loss function is non-convex, pathologically curved, and for which the Fisher matrix is known analytically (Bernacchia et al., 2018, Huh, 2020). The results presented in Fig. 1 were obtained with a network of 20 (linear) layers of size $n = 20$ each. We generated data by instantiating a teacher network with random Gaussian weights $\mathcal{N}(0, 1/n)$ in each layer and a predictive density with mean equal to the activation of the last layer and isotropic Gaussian noise with variance $\sigma_0^2 = 0.001^2$. Input samples were drawn from a standard normal distribution, and the teacher's predictive distribution was sampled accordingly to obtain output labels. We investigated the behaviour of FishLeg both in the large data regime (4000 training samples) and the small data regime (40 training samples).

In both regimes, FishLeg very rapidly drove the training loss to the minimum set by the noise ($\sigma_0$), a minimum which was attained by neither manually-tuned SGD-m nor Adam (Fig. 1A). This is consistent with previous such comparisons in similar settings (Bernacchia et al., 2018). In the small data regime, FishLeg displays some overfitting but continues to compare favourably against first-order methods on test error. In wall-clock time, FishLeg was about 5 times slower than SGDm per training iteration in this case (Fig. 1C), but we show that a significant speed up can be obtained by updating $\lambda$ every 10 iterations at little performance cost (see Fig. 3).

We used this simple experiment to assess the effectiveness of auxiliary loss optimization during the course of training. The lower the auxiliary loss, the closer it is from the Legendre conjugate of the cross-entropy at $\mathbf{g}$, and therefore the closer $Q(\lambda)\mathbf{g}$ is from the natural gradient. We assessed how much progress was made on the auxiliary loss, relative to how it would evolve if $\lambda$ was held constant, as $\theta$ is being optimized (Fig. 1B). This revealed two distinct phases of learning. At first, $\theta$ (and

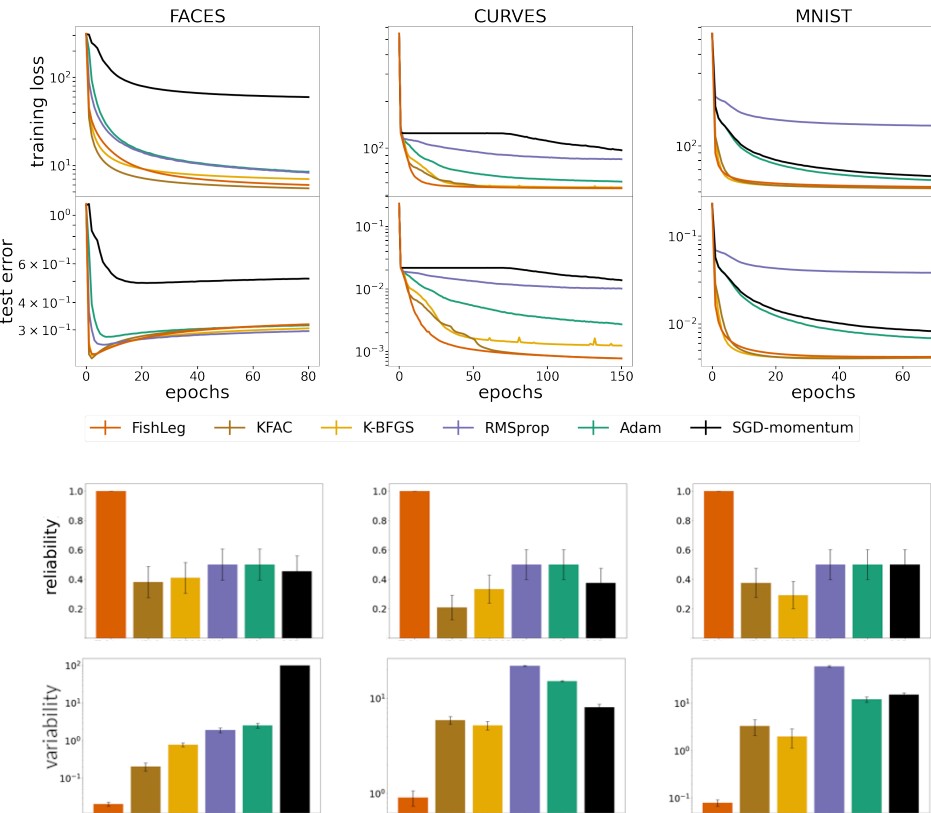

Figure 2: **Comparison between FishLeg and other optimizers** on standard auto-encoders benchmarks (batch size 100). **Top**: Training loss and test error as a function of training epochs on the 3 datasets, averaged over 10 different seeds. **Middle**: reliability of FishLeg across runs. For each optimizer and dataset, the fraction of runs (estimated from 10 runs) that converged properly (i.e. where the loss did not run out to 'nan') is shown; error bars show $\pm 1$ s.e.m. **Bottom**: variability (standard deviation) of the training loss across runs, averaged over epochs (discarding any epochs where the loss might have exploded). See supplementary Fig. 5 for results on batch size 1000.

therefore the FIM) changes rapidly, but $\boldsymbol{\lambda}$ rapidly adapts to keep the auxiliary loss in check (compare black and dark purple curves) – we speculate that this mostly reflects adaptation to the overall scale of the (inverse) FIM. Later on, as $\theta$ changes less rapidly, $\boldsymbol{\lambda}$ begins to learn the more subtle geometry of the FIM, giving rise to rather modest improvements on the auxiliary loss (compare black and lighter purple curves).

## 5.2 SECOND-ORDER OPTIMIZATION BENCHMARK

We applied FishLeg to the auto-encoders benchmarks previously used to compare second-order optimization methods – the details of these experiments (model architectures, datasets, etc) can be found in (Goldfarb et al., 2020), and hyperparameters specific to FishLeg are Table 1 (Appendix). To compare FishLeg to other optimizers, we used the code provided by Goldfarb et al. which implemented Adam, RMSprop, KFAC and KBFGS.

These autoencoder benchmarks are difficult problems on which the family of second-order methods has been shown to improve substantially over first-order optimizers (e.g. SGDm) and optimizers that exploit second-order information in diagonal form (Adam, RMSprop). Yet, no clear differences in performance has emerged *within* second-order methods. As far as training loss and test errors are concerned, our results confirmed this trend: FishLeg performed similarly to KFAC and KBFGS on the FACES and MNIST datasets, although it did converge within fewer iterations on the CURVES dataset (Fig. 2, top).

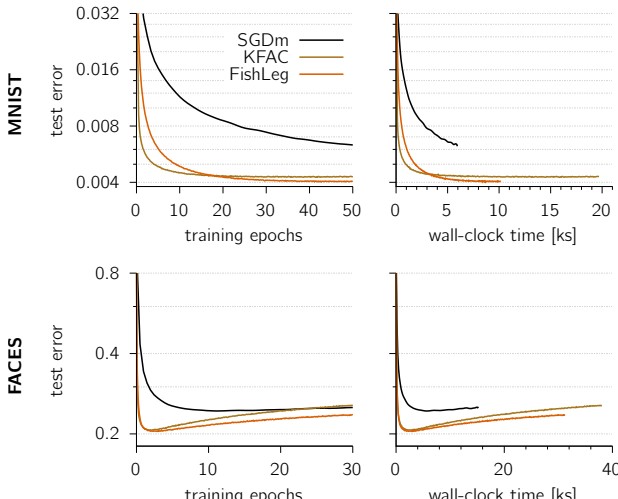

Figure 3: Wall-clock time comparisons on MNIST and FACES, with optimizers all implemented in the same way on CPU (Intel Xeon Platinum 8380H @ 2.90GHz) with OpenBLAS compiled for that architecture and multi-threaded with OpenMP (8 threads). Within each dataset, all curves show the same number of epochs to facilitate comparisons of CPU time-per-iteration. In FishLeg, the auxiliary parameters were updated only every 10 iterations, with no noticeable drop in performance but large wall-clock speedup. Similarly, KFAC's preconditioning matrices were inverted at each layer every 20 iterations only.

Interestingly, however, we found FishLeg to be consistently more robust than other methods, especially when using relatively small batch sizes. In particular, it converged more reliably over repeated runs with different random initializations (Fig. 2, middle). This is perhaps due to other methods achieving their best performance in a near-critical hyperparameter regime in which optimization tends to fail half of the time (these failed runs are often discarded when averaging). In contrast, FishLeg achieved similar or better performance in a more stable hyperparameter regime, and did not fail a single of the 10 runs we performed. Similarly, FishLeg was also more consistent than other methods, i.e. displayed a lower variance in training loss across runs (Fig. 2, bottom).

Given the heterogeneity of software systems in which the various methods were implemented (e.g. JAX vs PyTorch), we ran a clean wallclock-time comparison between SGDm, KFAC and FishLeg using a unified CPU-only implementation applied to the FACES and MNIST benchmarks. This ensured e.g. that the loss and its gradients were computed in exactly the same way across methods. Overall, one iteration of vanilla FishLeg was $\sim 5$ times slower than one iteration of SGDm. However, we were able to bring this down to only twice slower by updating $\lambda$ every 10 iterations, which did not significantly affect performance. Combined with FishLeg's faster progress per-iteration, this meant that FishLeg retained a significant advantage in wall-clock time over SGD (Fig.3), similar to KFAC. In practice we think that it might make sense to update $\lambda$ more frequently at the beginning of training, and let these updates become sparser as optimization progresses.

## 6  DISCUSSION

We provided a general framework for approximating the natural gradient through online meta-learning of the LF conjugate of a specific cross-entropy. Our framework is general: different choices can be made for how to meta-learn the LF conjugate (Eq. 11), parameterize its gradient (Eq. 14) and evaluate/differentiate the auxiliary loss (Eq. 15). Beyond our specific implementation, future work will study alternative choices that may be more efficient. For example, implicit differentiation (Lorraine et al., 2020, Rajeswaran et al., 2019), or evolution strategies (Metz et al., 2019, Gao & Sener, 2022) may be used to meta-learn the LF conjugate. The auxiliary loss could also be evaluated without taking the small $\epsilon$ limit but using antithetic estimators to reduce variance (Gao & Sener, 2022). Alternative parameterizations could be used for $\bar{\delta}$ or $Q$. Indeed, preliminary results show that more expressive choices for $Q$ yield better approximations of the inverse FIM (Appendix A.6).

Previous work on natural gradient aimed at computing and inverting analytically an approximation to the FIM, which was done on a case-by-case basis for dense (Martens & Grosse, 2015) and convolutional (Grosse & Martens, 2016) layers, and standard recurrent networks (Martens et al., 2018). By parameterizing the inverse FIM directly, our approach allows the user to express their assumptions about the structure of parameter *precisions*, which is typically easier to reason about than the structure of parameter *covariances*. We therefore expect that FishLeg will facilitate future applications of natural gradient optimization to a broader range of network architectures.

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

# A   APPENDIX

## A.1   ALGORITHM

---

**Algorithm 1** FishLeg algorithm (online setting)

---

1: **function** UPDATE_AUX($\boldsymbol{\theta}$, **g**, $\boldsymbol{\lambda}$, Adam_state)
2:    $\tilde{\mathbf{g}} \leftarrow \mathbf{g}/\|\mathbf{g}\|$                                                                     ▷ *normalize gradient*
3:    initialize the adjoint of $\boldsymbol{\lambda}$ to prepare for automatic differentiation reverse pass (c.f. line 9).
4:    $\bar{\boldsymbol{\delta}} \leftarrow Q(\boldsymbol{\lambda})\tilde{\mathbf{g}}$                              ▷ *can exploit fast matrix-vector products without forming Q*
5:    ▷ *Hessian-vector product on the regularized cross-entropy $\mathcal{H}$ (with $\bar{\boldsymbol{\delta}}$ taken off the automatic differentiation tape!) evaluated on a mini-batch different from the one used to obtain* **g**.                                                                                                                ◁
6:    $\mathbf{h} \leftarrow$ Hess_vec_prod(fun $\boldsymbol{\delta} \mapsto \mathcal{H}(\boldsymbol{\theta}, \boldsymbol{\delta})$, at $\boldsymbol{\delta} = 0$, along $\mathbf{v} =$ stop_gradient($\bar{\boldsymbol{\delta}}$))
7:    $\mathbf{d} \leftarrow \mathbf{h} - \tilde{\mathbf{g}}$
8:    aux_loss $\leftarrow \frac{1}{2}(d - \tilde{g})^{\top}$ stop_gradient($\bar{\boldsymbol{\delta}}$)               ▷ *log the value of the auxiliary loss*
9:    $\Delta_{\lambda} \leftarrow$ adjoint of $\boldsymbol{\lambda}$ after completion of reverse pass on surrogate auxiliary loss $d^{\top}\bar{\boldsymbol{\delta}}$
10:    $\boldsymbol{\lambda},$ Adam_state $\leftarrow$ Adam_update($\boldsymbol{\lambda}, \Delta_{\lambda}, \alpha,$ Adam_state)               ▷ $\alpha$ *is the learning rate*
11:    **return** aux_loss, $\lambda$, Adam_state
12:
13: $t \leftarrow 0$, initialize $\boldsymbol{\theta}_0$ and $\boldsymbol{\lambda}_0$
14: **while** not converged **do**
15:    $\mathcal{L}, \mathbf{g} \leftarrow$ value and gradient of negative log likelihood evaluated at $\boldsymbol{\theta}_t$ on a minibatch
16:    $\bar{\mathbf{g}} \leftarrow \beta\bar{\mathbf{g}} + (1 - \beta)\mathbf{g}$                                                           ▷ *momentum*
17:    ▷ *Note: the following step needs not be performed at every iteration – wall-clock time speedups can be obtained by running this every 10 iterations, though this might depend on the problem.*                                                                                                                                ◁
18:    aux_loss, $\boldsymbol{\lambda}_{t+1}$, Adam_state $\leftarrow$ UPDATE_AUX($\boldsymbol{\theta}_t, \bar{\mathbf{g}}, \boldsymbol{\lambda}_t,$ Adam_state)
19:    $\boldsymbol{\theta}_{t+1} \leftarrow \boldsymbol{\theta}_t - \eta Q(\boldsymbol{\lambda}_{t+1})\bar{\mathbf{g}}$
20:    $t \leftarrow t + 1$

---

## A.2   PROOF OF THEOREM 1

We start by stating the following Lemma, which is a modified version of Gibbs' inequality (see e.g. Chapter 2.6 of MacKay et al., 2003)

**Lemma 1.** *For a fixed $\boldsymbol{\theta}$, the regularized cross entropy between $p(\mathcal{D}|\boldsymbol{\theta})$ and $p(\mathcal{D}|\boldsymbol{\theta} + \boldsymbol{\delta})$*

$$\mathcal{H}_{\gamma}(\boldsymbol{\theta}, \boldsymbol{\delta}) = \mathbb{E}_{\mathcal{D} \sim p(\mathcal{D}|\boldsymbol{\theta})} \ell(\boldsymbol{\theta} + \boldsymbol{\delta}, \mathcal{D}) + \frac{\gamma}{2}\|\boldsymbol{\delta}\|^2 \tag{22}$$

*has a unique global minimum at $\boldsymbol{\delta} = \mathbf{0}$.*

*Proof of Theorem 1.* We start by computing the gradient and Hessian of the regularized cross-entropy $\mathcal{H}_{\gamma}(\boldsymbol{\theta}, \boldsymbol{\delta})$ with respect to $\boldsymbol{\delta}$, computed at $\boldsymbol{\delta} = 0$.

$$\nabla_{\boldsymbol{\delta}}\mathcal{H}_{\gamma}(\boldsymbol{\theta}, \mathbf{0}) = \mathbb{E}_{\mathcal{D} \sim p(\mathcal{D}|\boldsymbol{\theta})} \nabla_{\boldsymbol{\theta}}\ell(\boldsymbol{\theta}, \mathcal{D}) + \mathbf{0} = \mathbf{0} \tag{23}$$

$$\nabla_{\boldsymbol{\delta}}^2\mathcal{H}_{\gamma}(\boldsymbol{\theta}, \mathbf{0}) = \mathbb{E}_{\mathcal{D} \sim \mathbf{p}(\mathcal{D}|\boldsymbol{\theta})} \nabla_{\boldsymbol{\theta}}^2\ell(\boldsymbol{\theta}, \mathcal{D}) + \gamma\mathbf{I} = \mathcal{I}(\boldsymbol{\theta}) + \gamma\mathbf{I}. \tag{24}$$

The gradient is equal to the expectation of the score function, which is known to be equal to zero (as also implied by Lemma 1). The Hessian is equal to the damped FIM, as expressed by equation 2. Note that the damped FIM is positive definite and so its inverse exists.

Using property 9 of the LF conjugate, we express the *inverse* damped Fisher matrix as the Hessian of the LF conjugate of $\mathcal{H}_{\gamma}$. We denote by $\mathcal{H}_{\gamma}^{\star}(\boldsymbol{\theta}, \mathbf{u})$ the LF conjugate of $\mathcal{H}_{\gamma}(\boldsymbol{\theta}, \boldsymbol{\delta})$, as a function of its second argument $\boldsymbol{\delta}$. Using the definition of LF conjugate 6, that is equal to

$$\mathcal{H}_{\gamma}^{\star}(\boldsymbol{\theta}, \mathbf{u}) = \min_{\boldsymbol{\delta}} \mathcal{H}_{\gamma}(\boldsymbol{\theta}, \boldsymbol{\delta}) - \mathbf{u}^T\boldsymbol{\delta} \tag{25}$$

We denote by $\tilde{\boldsymbol{\delta}}_{\gamma}(\boldsymbol{\theta}, \mathbf{u})$ the minimizer of this expression, i.e.

$$\tilde{\boldsymbol{\delta}}_{\gamma}(\boldsymbol{\theta}, \mathbf{u}) = \text{argmin}_{\boldsymbol{\delta}} \mathcal{H}_{\gamma}(\boldsymbol{\theta}, \boldsymbol{\delta}) - \mathbf{u}^T\boldsymbol{\delta} \tag{26}$$

Using the property of LF conjugate, equation 9, we have that

$$\nabla_{\mathbf{u}}^2 \mathcal{H}_\gamma^\star(\boldsymbol{\theta}, \mathbf{u}) = \left( \nabla_{\boldsymbol{\delta}}^2 \mathcal{H}_\gamma(\boldsymbol{\theta}, \tilde{\boldsymbol{\delta}}_\gamma(\boldsymbol{\theta}, \mathbf{u})) \right)^{-1} \tag{27}$$

Comparing with the expression 24 of the damped FIM, the right hand side of this equation is equal to the inverse damped FIM when $\tilde{\boldsymbol{\delta}}_\gamma(\boldsymbol{\theta}, \mathbf{u})$ is equal to zero. By Lemma 1, we have that $\tilde{\boldsymbol{\delta}}_\gamma(\boldsymbol{\theta}, \mathbf{0}) = \mathbf{0}$ and therefore

$$(\mathcal{I}(\boldsymbol{\theta}) + \gamma I)^{-1} = \nabla_{\mathbf{u}}^2 \mathcal{H}_\gamma^\star(\boldsymbol{\theta}, \mathbf{0}). \tag{28}$$

Finally, using the properties of LF conjugate (Eq. 7), we have that $\tilde{\boldsymbol{\delta}}_\gamma(\boldsymbol{\theta}, \mathbf{u}) = \nabla_{\mathbf{u}} \mathcal{H}_\gamma^\star(\boldsymbol{\theta}, \mathbf{u})$ and the inverse FIM is equal to

$$(\mathcal{I}(\boldsymbol{\theta}) + \gamma I)^{-1} = \nabla_{\mathbf{u}} \tilde{\boldsymbol{\delta}}_\gamma(\boldsymbol{\theta}, \mathbf{0}). \tag{29}$$

$\square$

### A.3 INFORMAL ARGUMENT FOR THEOREM 1

Theorem 1 suggests that computation of the inverse damped FIM requires computing $\tilde{\boldsymbol{\delta}}_\gamma(\boldsymbol{\theta}, \mathbf{u})$ near $\mathbf{u} = \mathbf{0}$. By Lemma 1, we have that $\tilde{\boldsymbol{\delta}}_\gamma(\boldsymbol{\theta}, \mathbf{0}) = \mathbf{0}$, therefore we may hypothesize that $\tilde{\boldsymbol{\delta}}(\boldsymbol{\theta}, \mathbf{u})$ is near zero when $\mathbf{u}$ is also near zero. Under this assumption, and using equations 23, 24, we may approximate the regularized cross entropy $\mathcal{H}_\gamma(\boldsymbol{\theta}, \boldsymbol{\delta})$ by a second order Taylor expansion in $\boldsymbol{\delta}$:

$$\mathcal{H}_\gamma(\boldsymbol{\theta}, \boldsymbol{\delta}) - \mathbf{u}^T \boldsymbol{\delta} \simeq \mathcal{H}_\gamma(\boldsymbol{\theta}, \mathbf{0}) + \frac{1}{2} \boldsymbol{\delta}^T (\mathcal{I}(\boldsymbol{\theta}) + \gamma I) \boldsymbol{\delta} - \mathbf{u}^T \boldsymbol{\delta} \tag{30}$$

Minimizing this expression with respect to $\boldsymbol{\delta}$ results in

$$\tilde{\boldsymbol{\delta}}_\gamma(\boldsymbol{\theta}, \mathbf{u}) = (\mathcal{I}(\boldsymbol{\theta}) + \gamma I)^{-1} \mathbf{u} \tag{31}$$

which implies the statement of the theorem, $(\mathcal{I}(\boldsymbol{\theta}) + \gamma I)^{-1} = \nabla_{\mathbf{u}} \tilde{\boldsymbol{\delta}}_\gamma(\boldsymbol{\theta}, \mathbf{0})$.

### A.4 PROOFS OF CONVERGENCE

**Definition 1** (PL Condition). *We say that a function satisfies the PL condition with parameter $\mu \in \mathbb{R}_{>0}$, if the following holds*

$$\|\nabla \mathcal{L}(\boldsymbol{\theta})\|^2 \geq \mu(\mathcal{L}(\boldsymbol{\theta}) - \mathcal{L}(\boldsymbol{\theta}^\star)), \quad \forall \boldsymbol{\theta} \tag{32}$$

*We use the notation $\mu$-PL to denote the class of functions satisfying the PL condition with parameter $\mu$.*

This condition implies that every stationary point is a global minimum. It however does not imply either uniqueness of the global minimum nor convexity.

**Definition 2** (Smoothness). *We say that a function is smooth with parameter $\xi$, if the following holds*

$$\mathcal{L}(\boldsymbol{\theta}') \leq \mathcal{L}(\boldsymbol{\theta}) + \nabla \mathcal{L}(\boldsymbol{\theta})^T (\boldsymbol{\theta}' - \boldsymbol{\theta}) + \frac{\xi}{2} \|\boldsymbol{\theta}' - \boldsymbol{\theta}\|^2, \quad \forall \boldsymbol{\theta}, \boldsymbol{\theta}' \tag{33}$$

*We use the notation $\xi$-smooth to denote the class of smooth functions with parameter $\xi$.*

*Proof of theorem 2.* We show that the error at each iteration $t + 1$ is linearly related to the error at iteration $t$.

$$\mathcal{L}(\boldsymbol{\theta}_{t+1}) - \mathcal{L}(\boldsymbol{\theta}_t)$$

$$\leq \quad -\eta \mathbf{g}(\boldsymbol{\theta}_t)^\top Q \mathbf{g}(\boldsymbol{\theta}_t) + \frac{\xi \eta^2}{2} \mathbf{g}(\boldsymbol{\theta}_t)^\top Q^2 \mathbf{g}(\boldsymbol{\theta}_t) \tag{34}$$

$$\leq \quad -\eta \mathbf{g}(\boldsymbol{\theta}_t)^\top Q \mathbf{g}(\boldsymbol{\theta}_t) + \frac{\xi \lambda_{\max} \eta^2}{2} \mathbf{g}(\boldsymbol{\theta}_t)^\top Q \mathbf{g}(\boldsymbol{\theta}_t) \tag{35}$$

$$= \quad -\eta (1 - \frac{\xi \lambda_{\max} \eta}{2}) \mathbf{g}(\boldsymbol{\theta}_t)^\top Q \mathbf{g}(\boldsymbol{\theta}_t) \tag{36}$$

$$\leq \quad -\frac{\eta \lambda_{\min}}{2} \|\mathbf{g}(\boldsymbol{\theta}_t)\|^2 \tag{37}$$

$$\leq \quad -\frac{\eta \mu \lambda_{\min}}{2} (\mathcal{L}(\boldsymbol{\theta}_t) - \mathcal{L}(\boldsymbol{\theta}^*)) \tag{38}$$

$$= \quad -\frac{\mu \lambda_{\min}}{2 \xi \lambda_{\max}} (\mathcal{L}(\boldsymbol{\theta}_t) - \mathcal{L}(\boldsymbol{\theta}^*)) \tag{39}$$

The first line is a result of smoothness assumption and the update rule 17. The second line follows from the definition of $\lambda_{\max}$. The third line is only a rearrangement of terms. The fourth line follows from the definition of $\lambda_{\min}$ and the choice of $\eta = \frac{1}{\xi \lambda_{\max}}$. The fifth line holds by PL condition. The last line follows from replacing the value of $\eta = \frac{1}{\xi \lambda_{\max}}$.

We thus have

$$\mathcal{L}(\boldsymbol{\theta}_{t+1}) - \mathcal{L}(\boldsymbol{\theta}^*) \leq (1 - \frac{\mu \lambda_{\min}}{2 \xi \lambda_{\max}}) \left( \mathcal{L}(\boldsymbol{\theta}_t) - \mathcal{L}(\boldsymbol{\theta}^*) \right).$$

Applying this recursive relation over $t$, starting from $\boldsymbol{\theta}_0$, we arrive at

$$\mathcal{L}(\boldsymbol{\theta}_t) - \mathcal{L}(\boldsymbol{\theta}^*) \leq (1 - \frac{\mu \lambda_{\min}}{2 \xi \lambda_{\max}})^t \left( \mathcal{L}(\boldsymbol{\theta}_0) - \mathcal{L}(\boldsymbol{\theta}^*) \right).$$

$\square$

*Proof of theorem 3.* We show that the error at each iteration $t + 1$ is related to the error at iteration $t$ as follows.

$$\mathbb{E}\left[ \mathcal{L}(\boldsymbol{\theta}_{t+1}) - \mathcal{L}(\boldsymbol{\theta}_t) | \boldsymbol{\theta}_t \right]$$

$$\leq \quad \mathbb{E}\left[ -\eta_t \mathbf{g}(\boldsymbol{\theta}_t)^\top Q \hat{\mathbf{g}}(\boldsymbol{\theta}_t) + \frac{\xi \eta_t^2}{2} \hat{\mathbf{g}}(\boldsymbol{\theta}_t)^\top Q^2 \hat{\mathbf{g}}(\boldsymbol{\theta}_t) | \boldsymbol{\theta}_t \right] \tag{40}$$

$$\leq \quad -\eta_t \mathbf{g}(\boldsymbol{\theta}_t)^\top Q \mathbf{g}(\boldsymbol{\theta}_t) + \mathbb{E}\left[ \frac{\xi \lambda_{\max}^2 \eta_t^2}{2} \|\hat{\mathbf{g}}(\boldsymbol{\theta}_t)\|^2 | \boldsymbol{\theta}_t \right] \tag{41}$$

$$\leq \quad -\eta_t \lambda_{\min} \|\mathbf{g}(\boldsymbol{\theta}_t)\|^2 + \frac{\xi \eta_t^2 \lambda_{\max}^2 G^2}{2} \tag{42}$$

$$\leq \quad -\eta_t \mu \lambda_{\min} (\mathcal{L}(\boldsymbol{\theta}_t) - \mathcal{L}(\boldsymbol{\theta}^*)) + \frac{\xi \eta_t^2 \lambda_{\max}^2 G^2}{2} \tag{43}$$

The first line is a result of smoothness assumption and the update rule 17. The second line follows from the definition of $\lambda_{\max}$. The third line follows from the upper bound assumption on the norm of gradient. The fourth line holds by PL condition.

We thus have

$$\mathcal{L}(\boldsymbol{\theta}_{t+1}) - \mathcal{L}(\boldsymbol{\theta}^*) \leq (1 - \eta_t \mu \lambda_{\min})(\mathcal{L}(\boldsymbol{\theta}_t) - \mathcal{L}(\boldsymbol{\theta}^*)) + \frac{\xi \eta_t^2 \lambda_{\max}^2 G^2}{2}$$

Recall the choice of $\eta_t = \frac{2}{\mu\lambda_{\min}(t+1)}$. By definition of $A$, we have $\mathcal{L}(\boldsymbol{\theta}_t) - \mathcal{L}(\boldsymbol{\theta}^*) \leq \frac{A}{t+1}$ when $t = 0$. Using induction, we prove the same for $\mathcal{L}(\boldsymbol{\theta}_{t+1}) - \mathcal{L}(\boldsymbol{\theta}^*)$.

$$\mathcal{L}(\boldsymbol{\theta}_{t+1}) - \mathcal{L}(\boldsymbol{\theta}^*) \leq (1 - \frac{2}{t+1})\frac{A}{(t+1)} + \frac{2\xi\lambda_{\max}^2 G^2}{\mu^2\lambda_{\min}^2(t+1)^2} \tag{44}$$

$$\leq \frac{A}{t+1} - \frac{A}{(t+1)^2} \tag{45}$$

$$= A(\frac{1}{t+1} - \frac{1}{(t+1)^2}) \tag{46}$$

$$\leq \frac{A}{t+2}. \tag{47}$$

That completes the proof.

$\square$

## A.5 PARAMETERIZATION OF THE MATRIX Q

Consider a multi-layer perceptron (MLP) neural network with $L$ layers. The activation $a_i^\ell$ of neuron $i$ at layer $\ell$ is equal to

$$a_i^\ell = \sigma_\ell\left(\sum_{j=1}^{N_{\ell-1}+1} W_{ij}^\ell a_j^{\ell-1}\right) \quad \text{for } 1 \leq i \leq N_\ell \tag{48}$$

$$\tag{49}$$

where layer $\ell$ has $N_\ell$ neurons and $\ell = 1, \ldots, L$. The symbol $W^\ell$ denotes the weight matrix from layer $\ell - 1$ to layer $\ell$, and has size $N_\ell \times (N_{\ell-1} + 1)$. This weight matrix includes a bias, by setting the $N_\ell + 1$ activation at each layer equal to one:

$$a_{N_\ell+1}^\ell = 1 \tag{50}$$

The function $\sigma_\ell$ can be any nonlinearity such as ReLU or a Softmax in case of the last layer. The output of the neural network is defined as the activation in the last layer, $a^L$. For convenience of notation, the input to the neural network, denoted by $x$ and of dimension $N_0$, is defined as activation at layer 0:

$$a_i^0 = x_i \quad \text{for } 1 \leq i \leq N_0 \tag{51}$$

$$a_{N_0+1}^0 = 1 \tag{52}$$

We structure the matrix $Q$ as block-diagonal, each block corresponds to one layer and has the same number of rows and columns, equal to $N_\ell(N_{\ell-1}+1)$. For layer $l$, We parameterize the corresponding block, denoted by $Q_l$ as

$$Q_\ell = (R_\ell R_\ell^T \otimes L_\ell L_\ell^T) \tag{53}$$

where the matrix $R_\ell$ has size $(N_{\ell-1} + 1) \times (N_{\ell-1} + 1)$ while the matrix $L_\ell$ has size $N_\ell \times N_\ell$. Both matrices $L_\ell, R_\ell$ are lower triangular. This parameterization ensures that the matrix $Q$ is positive definite. The auxiliary parameters $\boldsymbol{\lambda}$ are represented by the matrices $L_\ell, R_\ell$ for all layers $\ell = 1, \ldots, L$, for a total of $\frac{1}{2}\sum_{\ell=1}^L [N_\ell(N_\ell + 1) + (N_{\ell-1} + 1)(N_{\ell-1} + 2)]$ auxiliary parameters. This number is much smaller than the total number of entries of the matrix $Q$, which is equal to $\left(\sum_{\ell=1}^L N_\ell(N_{\ell-1} + 1)\right)^2$

## A.6 PRELIMINARY INVESTIGATIONS OF MORE FLEXIBLE APPROXIMATIONS OF THE INVERSE FIM

Directly parameterizing and learning the inverse FIM lends more flexibility than parameterizing and learning the FIM in an easy-to-invert form. We conducted preliminary experiments with alternative parameterizations $Q(\boldsymbol{\lambda})$ of the inverse FIM (i.e. beyond the one described in Appendix A.5), all constrained to afford fast $Q(\boldsymbol{\lambda})\mathbf{g}$ products. In particular, we experimented with:

- a simple diagonal matrix $Q$ (constrained to be positive definite), mostly included as a useful baseline

- the block Kronecker used in the paper and described in Appendix A.5, known to perform much better than a diagonal matrix;

- a modification of the block Kronecker form that introduces full inner and outer diagonal rescaling: at each layer $\ell$, we took

$$Q_\ell = A_\ell(R_\ell \otimes L_\ell)B_\ell^2(R_\ell^T \otimes L_\ell^T)A_\ell$$

where $A_\ell$ and $B_\ell$ are two diagonal matrices of the appropriate size. The presence of $B_\ell$ makes this similar to EKFAC (George et al., 2018), but $A_\ell$ makes it even more general;

- a sum of block Kronecker approximations: at each layer $\ell$, we took

$$Q_\ell = (R_\ell^{(1)}R_\ell^{(1)^T}) \otimes (L_\ell^{(1)}L_\ell^{(1)^T}) + (R_\ell^{(2)}R_\ell^{(2)^T}) \otimes (L_\ell^{(2)}L_\ell^{(2)^T})$$

All these approximations lend themselves to efficient $Q\mathbf{g}$ products (e.g. by exploiting standard properties of the Kronecker product).

To assess how well these approximations could approximate the inverse FIM at a point in optimization where high-quality curvature information matters, we pre-trained an MLP with layer sizes $[784, 300, 100, 30, 100, 300, 784]$ for 200 iterations on MNIST autoencoding using SGD-with-momentum. The resulting model parameters $\boldsymbol{\theta}$ were subsequently frozen, and the auxiliary loss was minimized to convergence (asymptotic value shown here) under each of the four different parameterizations of $Q(\boldsymbol{\lambda})$ described above. We found that diagonal rescaling improved only slightly over the vanilla block Kronecker approximation. However, there was a larger improvement going from the block Kronecker approximation to the sum-of-block-Kronecker approximation – indeed, the improvement on the auxiliary loss was a sizeable fraction of the improvement going from a diagonal approximation to single block-Kronecker terms. The latter is known to be highly consequential for training, and so we speculate that sum-of-Kronecker approximations might improve natural gradient descent in future work.

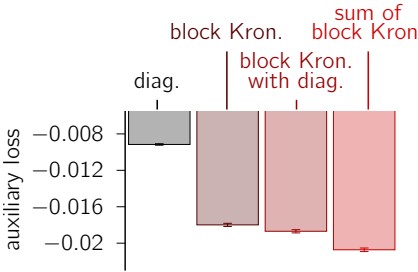

Figure 4: **Comparison of various forms of $Q(\boldsymbol{\lambda})$.** See text in Appendix A.6 for details.

### A.7 FISHLEG COMPLEXITY

In this section, we provide an analysis of the complexity of FishLeg for a deep fully connected network with $L$ layers of size $N$ each, processing data in mini-batches of size $K$. We assume that $Q$ has the block-diagonal Kronecker form of Eq. 53. The complexity of the forward pass is $\mathcal{O}(LKN^2)$, and with automatic differentiation the backward pass has the same complexity. At each layer, the gradient of the loss w.r.t. the weight matrix has rank $R = \min(N, K)$, such that computing the $Q\boldsymbol{g}$ product required for the FishLeg update step (Eq. 17) using Kronecker identities has complexity $\mathcal{O}(LKN^2)$ or $\mathcal{O}(LN^3)$, whichever is smallest. Finally, the inner update step (Eq. 16) is dominated by (i) the $Q\tilde{\boldsymbol{g}}$ product (line 4 of Algorithm 1) and (ii) the Hessian-vector product (line 6) which has the same $\mathcal{O}(LKN^2)$ complexity as the forward pass. Altogether, FishLeg has complexity $\mathcal{O}(LKN^2)$.

In contrast, KFAC has complexity $\mathcal{O}(LN^2 \max(K, N))$, as the $N^3$ cost of inverting the Kronecker factors is unavoidable even when the batch size is small.

### A.8 FishLeg as amortization of CG steps in Hessian-free optimization

FishLeg can be thought as a way of gradually amortizing the conjugate gradients (CG)-based inner loop in Hessian-free optimization. Looking back at Eq. 20, one might wonder: why not directly optimize over the product $\mathbf{v} = Q(\boldsymbol{\lambda})\mathbf{g}$, instead of optimizing over $\boldsymbol{\lambda}$? Treating $\mathbf{v}$ as free parameters, this would entail minimizing $\frac{1}{2}\mathbf{v}^\top(\mathcal{I}(\boldsymbol{\theta}) + \gamma I)\mathbf{v} - \mathbf{v}^\top\mathbf{g}$, which is in fact exactly what Hessian-free optimization does using CG (Byrd et al., 2011; Martens et al., 2010; though $\mathcal{I}$ is sometimes replaced by the Gauss-Newton approximation to the Hessian). Although each step of CG has a cost comparable to the computation of $\mathbf{g}$, in practice several iterations are required which tends to remove any advantage on wallclock time. By learning $Q(\boldsymbol{\lambda})$ instead of $Q(\boldsymbol{\lambda})\mathbf{g}$, FishLeg learns to amortize those CG steps over a progressively growing subspace of noisy gradients.

### A.9 Analytical results in a simple linear-Gaussian system

Here, we provide an analytical derivation of the behaviour of FishLeg in a simple one-layer linear network with a Gaussian likelihood, and draw some insights. Consider a regression model of the form $p(y|x) = \mathcal{N}(y; \boldsymbol{\theta}^T x, 1)$, and a teacher in the same model class with $\boldsymbol{\theta} = 0$. Let the input distribution be $x \sim \mathcal{N}(0, \Sigma)$. The log likelihood – in the limit of large data batches sampled from the teacher network – is given by $\ell(\boldsymbol{\theta}) = \frac{1}{2}\boldsymbol{\theta}^T\Sigma\boldsymbol{\theta}$. For analytical convenience, we will work in the continuous time limit. In this limit, the standard gradient flow is given by

$$\frac{d\boldsymbol{\theta}}{dt} = -\Sigma\boldsymbol{\theta}. \tag{54}$$

This flow is easily seen to converge to the correct solution (the teacher parameter $\boldsymbol{\theta} = 0$), but to do so slowly along the eigenvectors of $\Sigma$ associated with small eigenvalues. Indeed, writing $\boldsymbol{\theta} = U\boldsymbol{a}$ where the columns of $U$ are the orthonormal eigenvectors of $\Sigma$, Eq. 54 implies $a_i(t) = a_i(0)e^{-\lambda_i t}$ where $\lambda_i > 0$ is the $i^{\text{th}}$ eigenvalue of $\Sigma$. Thus, when $\Sigma$ is poorly conditioned, convergence is very slow in its bottom subspace. The question arises: how well does FishLeg mitigate this problem?

For this simple model, the Fisher matrix is $\mathcal{I} = \Sigma$, and the cross-entropy of Eq. 10 (unregularized, as it is not necessary here) is given by $\mathcal{H}(\cdot, \boldsymbol{\delta}) = \frac{1}{2}\boldsymbol{\delta}^T\Sigma\boldsymbol{\delta}$. To simplify our analytical derivations, instead of using Adam for optimizing the auxiliary loss as we do in our experiments, here we consider simple gradient descent in continuous time:

$$\frac{dQ}{dt} = -\alpha\nabla_Q\mathcal{A}\left(\boldsymbol{\theta}, \boldsymbol{u} = \frac{\Sigma\boldsymbol{\theta}}{\|\Sigma\boldsymbol{\theta}\|}, Q\right) \tag{55}$$

which is evaluated at $\boldsymbol{u} = \boldsymbol{g}/\|\boldsymbol{g}\|$ where $\boldsymbol{g} = \Sigma\boldsymbol{\theta}$ is the momentary gradient of the primary loss (the model's negative log likelihood). Expanding the auxiliary loss of Eq. 20, we obtain

$$\frac{dQ}{dt} = -\frac{\alpha}{\|\Sigma\boldsymbol{\theta}\|^2}\nabla_Q\left[\boldsymbol{\theta}^T\Sigma\left(\frac{1}{2}Q\Sigma Q - Q\right)\Sigma\boldsymbol{\theta}\right] \tag{56}$$

$$= -\frac{\alpha}{\|\Sigma\boldsymbol{\theta}\|^2}\nabla_Q\text{Trace}\left[\left(\frac{1}{2}Q\Sigma Q - Q\right)\Sigma\boldsymbol{\theta}\boldsymbol{\theta}^T\Sigma\right] \tag{57}$$

Letting $\boldsymbol{z} = Q\Sigma\boldsymbol{\theta}$, and after some algebra, the auxiliary flow of $Q$ in Eq. 57 implies the following flow for $\boldsymbol{z}$:

$$\frac{d\boldsymbol{z}}{dt} = \alpha\Sigma(\boldsymbol{\theta} - \boldsymbol{z}). \tag{58}$$

With the same notation, the primary FishLeg flow is

$$\frac{d\boldsymbol{\theta}}{dt} = -Q\Sigma\boldsymbol{\theta} = -\boldsymbol{z}. \tag{59}$$

Thus, in this simple linear-Gaussian setup, FishLeg boils down to a pair of coupled linear ODEs. One can already see that, assuming a separation of timescales with $\alpha \gg 1$ such that $\boldsymbol{\theta}$ changes much more slowly than $Q$ (i.e. than $\boldsymbol{z}$), then the FishLeg flow becomes $d\boldsymbol{\theta}/dt = -\boldsymbol{\theta}$ which implies exponential decay of the loss irrespective of $\Sigma$ – and indeed, for this model it is exactly natural gradient descent. To drive the point home, we rewrite these coupled ODEs in the orthonormal eigenbasis of $\Sigma = U\Lambda U^T$, which yields

$$\frac{d}{dt}\begin{pmatrix} \hat{\boldsymbol{\theta}} \\ \hat{\boldsymbol{z}} \end{pmatrix} = \begin{pmatrix} 0 & -I \\ \alpha\Lambda & -\alpha\Lambda \end{pmatrix}\begin{pmatrix} \hat{\boldsymbol{\theta}} \\ \hat{\boldsymbol{z}} \end{pmatrix} \tag{60}$$

where we have defined $\boldsymbol{\theta} = U\hat{\boldsymbol{\theta}}$ and $\boldsymbol{z} = U\hat{\boldsymbol{z}}$, and $\Lambda$ is a diagonal matrix containing the eigenvalues $\{\lambda_i\}$ of the input covariance. The eigenvalues $\{\beta_i\}$ of this state matrix are then easily shown to satisfy

$$\alpha\lambda_i(1 + \beta_i) = -\beta_i^2 \tag{61}$$

assuming the correct ordering of the $\beta_i$'s w.r.t. the $\lambda_i$'s. For each $\lambda_i$, as $\alpha$ grows large (i.e. good separation of timescales between $\boldsymbol{\theta}$ and $Q$), there are only two ways for $\beta_i$ to satisfy Eq. 61. Either $\beta_i$ remains finite ($\mathcal{O}(\alpha^0)$), implying that $(1 + \beta_i)$ must be $\mathcal{O}(1/\alpha)$, which in turn implies $\beta_i \rightarrow -1$. Or $\beta_i$ grows large and real, in which case $(1 + \beta_i)$ must be negative because $-\beta_i^2$ is, and more specifically $\beta_i$ must grow as $-\mathcal{O}(\alpha)$. These qualitative arguments can be confirmed by writing down the closed-form solution of the quadratic Eq. 61. In summary, with sufficient separation of timescales between the primary and auxiliary parameter updates, FishLeg converges uniformly in all directions in parameter space, with a uniform dominant timescale that does not depend on the condition number of $\Sigma$. This is contrast with standard gradient descent which is slowed down by small negative eigenvalues in the spectrum of $\Sigma$.

Note that in the above derivations, the transition from Eq. 57 to Eq. 58 made the implicit assumption that $Q$ was unconstrained – in particular, it could lose symmetry. In this case, the derivation shows that unless the auxiliary flow is substantially faster than the primary flow, FishLeg can suffer from oscillations / poor damping especially in the bottom subspace of $\Sigma$ (c.f. solutions of Eq. 61 for small $\alpha$). In practice however, one would normally write $Q = LL^T$ with a positivity constraint on the diagonal of $L$, and formulated the auxiliary flow in terms of $dL/dt$ instead. This would guarantee $Q \succ 0$ throughout optimization. Although analytical derivations become more tedious in this case, we speculate based on numerical simulations that this parameterization mitigates the problem of oscillatory optimization dynamics.

| | FACES | | CURVES | | MNIST | |
|---|---|---|---|---|---|---|
| | 100 | 1000 | 100 | 1000 | 100 | 1000 |
| Damping $\gamma$ | 1.0 | 1e-3 | 1e-2 | 1.0 | 3e-1 | 3e-2 |
| $\eta$ | 5e-2 | 5e-2 | 1e-2 | 1e-1 | 7e-2 | 5e-2 |
| $\alpha$ | 1e-4 | 1e-4 | 1e-5 | 5e-4 | 1e-4 | 1e-4 |
| $\eta_{\text{SGDm}}$ | 1e-2 | 1e-2 | 3e-2 | 1e-2 | 1e-2 | 1e-3 |

Table 1: Optimal hyperparameter values for FishLeg, identified as the result of a grid search over the space shown in Table 2. These hyperparameters were chosen to minimise the training loss.

| | FACES | | CURVES | | MNIST | |
|---|---|---|---|---|---|---|
| | 100 | 1000 | 100 | 1000 | 100 | 1000 |
| Damping $\gamma$ | 1e-2, 1e-1, 1.0, 10, 100 | 1e-3, 1e-2, 1e-1, 1.0, 10, 100 | 1e-4, 1e-3, 3e-2, 1e-2, 3e-1, 1.0, 10 | 1e-4, 1e-3, 1e-2, 1e-1, 1.0 | 1e-4, 1e-2, 1e-1, 3e-1, 1.0, 100 | 1e-4, 1e-2, 3e-2, 1.0, 100 |
| $\eta$ | 1e-3, 5e-3, 1e-2, 5e-2 | 5e-4, 5e-3, 1e-3, 1e-2, 5e-2 | 1e-3, 5e-3, 1e-2, 5e-2, 7e-2, 1e-1, 2e-1, 5e-1 | 1e-3, 1e-2, 2e-2, 5e-2, 1e-1 | 1e-3, 5e-3, 1e-2, 5e-2, 7e-2, 1e-1, 2e-1, 5e-1 | 1e-3, 5e-3, 1e-2, 2e-2, 5e-2, 7e-2, 1e-1, 5e-1 |
| $\alpha$ | 5e-5, 8e-5, 1e-4, 5e-4, 1e-3, 5e-3 | 5e-5, 1e-4, 5e-4, 1e-3, 1e-2, 1e-1 | 1e-5, 5e-5, 1e-4, 5e-4, 1e-3, 5e-3 | 1e-5, 1e-4, 5e-4, 1e-3, 1e-2 | 1e-5, 5e-5, 1e-4, 5e-4, 1e-3, 5e-3 | 5e-5, 1e-4, 5e-4, 1e-3 |

Table 2: Entire space of hyperparameters explored to optimize the training loss for each task. All combinations were explored, in a grid fashion, and each combination was run three times with different seeds. Finally, the combination of hyperparameters with the lowest training loss was selected.

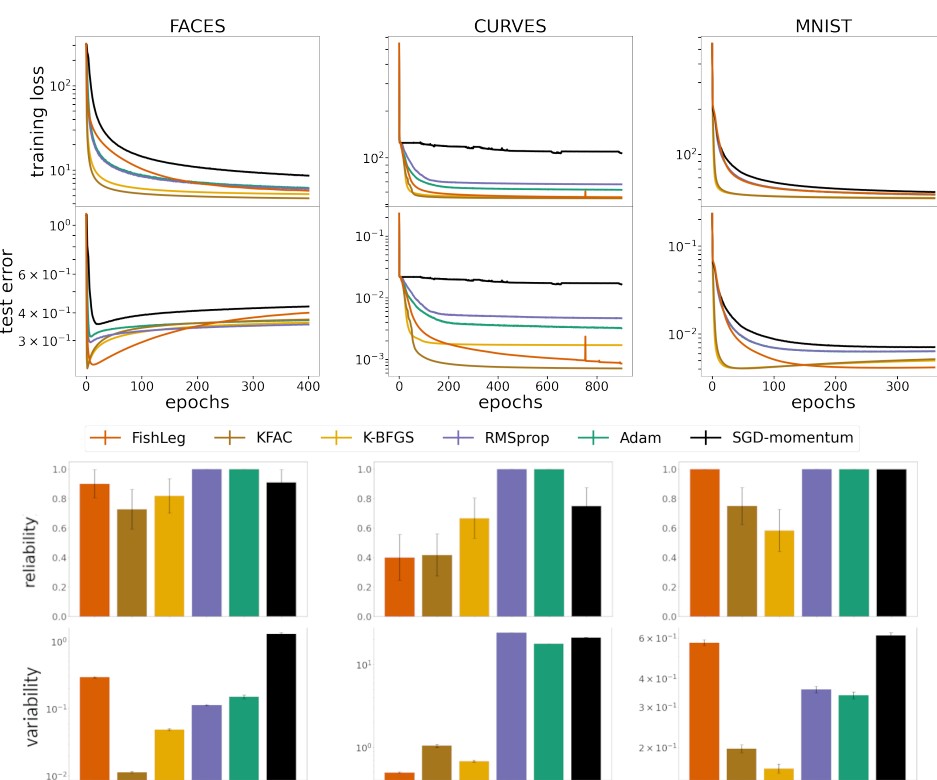

Figure 5: Same as Fig. 2 with a batch size of 1000.

