# OpenReview forum: "Fisher-Legendre (FishLeg) optimization of deep neural networks"
_ICLR.cc/2023/Conference — ICLR 2023 notable top 25%_

### Official Review · Reviewer_qrGz · 2022-10-23

**Confidence:** 5
**Correctness:** 4
**Technical Novelty And Significance:** 3
**Empirical Novelty And Significance:** 3
**Recommendation:** 8

**Clarity, Quality, Novelty And Reproducibility:**

Clarity: The paper is clear in both math and numerics.

Novelty: The usage of online learning and meta-learning in natural gradient is a new and good approach.

Reproducibility: Some analytical examples are needed.

**Strength And Weaknesses:**

Strength: The usage of Legendre and the auxiliary parameters plus meta-learning is an interesting idea. They have the ability to compute the inverse of the Fisher information matrix efficiently.

Weakness:

1. The authors clearly miss many important results on the approximation of the information matrix, using the Legendre duality and auxiliary variable.

Affine Natural Proximal Learning. Wasserstein Proximal of GANs, Wasserstein natural gradient, etc.

2. Does the authors have some analytical examples to demonstrate the effectiveness of algorithms? Some examples in Gaussian distributions could be useful.

**Summary Of The Paper:**

The authors propose a general method to approximate natural gradient updates in a deep-learning context. The natural gradient is a way of second-order optimization. They apply the Legendre-Fenchel duality to learn a direct and efficiently evaluated model for the product of the inverse Fisher with any vector. Under some assumptions, they prove that their approximate natural gradient descent can converge to a local minimum including a global minimum. Several numerical experiments are shown to demonstrate the effectiveness of the algorithm.

**Summary Of The Review:**

The paper is written well with clear mathematics and many numerical experiments. However, the authors still require some direct analytical examples in Gaussian distributions to demonstrate their ideas. Much important literature is missed in the context.

---

> ### Author Response · Authors · 2022-11-17
> **Example in the linear-Gaussian case**
>
> We thank the Reviewer for their time reviewing our paper, and for pointing out that Legendre duality has been used in the Wasserstein literature which we were not aware of! We are still diggesting these papers and thinking about how to best integrate them in our Background section ─ which we will do.
>
> We took the Reviewer's main comment to heart and are now showing analytical derivations in the Gaussian case in Appendix A.9. Specifically, we solve the behaviour of FishLeg analytically in a simple one-layer linear network with a Gaussian likelihood, and draw useful insights. We briefly summarize our approach and findings in this response post, but please refer to Appendix A.9 for the details.
>
> We considered a regression model of the form $p(y|x) = \mathcal{N}(y; \boldsymbol\theta^T x, 1)$, and a teacher in the same model class with $\boldsymbol\theta=0$ (WLOG). Let the input distribution be $x \sim \mathcal{N}(0,\Sigma)$. The log likelihood ─ in the limit of large data batches sampled from the teacher network ─ is given by $\ell(\boldsymbol\theta) = \frac12 \boldsymbol\theta^T \Sigma \boldsymbol\theta$. This is a simple quadratic loss, i.e. the simplest type of problem on which second-order methods can vastly outperform first-order ones. Working in continuous time for convenience, and using gradient descent instead of Adam for optimizing the auxiliary loss, we find that FishLeg boils down to a pair of coupled linear ODEs:
> \begin{equation}
>     \frac{d\boldsymbol\theta}{dt} = - \boldsymbol{z} \qquad \text{and} \qquad
>     \frac{d\boldsymbol{z}}{dt} = \alpha \Sigma (\boldsymbol\theta-\boldsymbol{z})
> \end{equation}
> where the parameter $\alpha$ expresses a separation of timescale between the primary and auxiliary flows.
> When the two timescales are well separated ($\alpha \gg 1$), the dominant eigenvalues of this linear system converge to -1, i.e. they lose their dependence on the eigenvalues of $\Sigma$, such that FishLeg becomes insensitive to potential ill-conditioning of $\Sigma$. In this limit, the FishLeg flow is exactly natural gradient descent, $d\boldsymbol\theta/dt = -\boldsymbol\theta$. This has been a useful exercise for us. In particular, along the way, it provided a neat justification for  evaluating our auxiliary loss and its gradients at the _normalized_ gradient $g/\|| g \||$ (c.f. Eq. 20) which we had found important to do based on our simulations.
>
> We hope this is what the Reviewer had in mind!

---

> > ### Comment · Reviewer_qrGz · 2022-11-23
> > **Reply to reviewers**
> >
> > The authors address my questions. This is a good paper. I will raise my score to 8.

---

### Official Review · Reviewer_QPAM · 2022-10-24

**Confidence:** 4
**Correctness:** 2
**Technical Novelty And Significance:** 2
**Empirical Novelty And Significance:** 2
**Recommendation:** 6

**Clarity, Quality, Novelty And Reproducibility:**

Clarity: The presentation can be improved.  Discuss the hidden assumptions and the hidden (time and space) cost of the proposed method.

Quality: The theoretical results and empirical results are not strong.

Novelty: It is interesting to use the positive-definite surrogate Q to approximate the Fisher matrix by introducing an additional cost.



**Strength And Weaknesses:**

Strength:
* Re-formulation of natural-gradient descent via the Legendre transformation.
* Surrogate approximation Q (Sec 4.2) to handle the positive-definite constraint in the Fisher information matrix.
* Some theoretical results of the proposed method.

Weakness:
---
* Theorem 2 and Theorem 3 are weak/standard since the authors make **direct** assumptions of the matrix Q. Note that Q is an approximation of the Fisher information matrix. Thus, the authors implicitly make assumptions about the Fisher information matrix.
Moreover, the impact of Eq 16 is unknown since the assumptions of the matrix Q bypass the convergence analysis of Eq 16 as shown in the proof of Thm 2 and 3.

---
* There is an implicit and key assumption in Theorem 1. Without addressing this issue, the theoretical results are pretty useless.
My main question is when/why  $- E_{p(\mathcal{D})}[ \nabla_\theta^2 \log p(\mathcal{D|\theta})] $ is positive-definite in NN cases, which often implies the NN is not over-parametrized. If $- E_{p(\mathcal{D})}[ \nabla_\theta^2 \log p(\mathcal{D|\theta})] $ is positive-definite, why is damping needed?  The authors should include a NN example to justify this assumption.

As mentioned in Eq (8), the H function must be **strictly convex** w.r.t. $\delta$.
As shown in Lemma 1 (in the appendix), the assumption $p(\mathcal{D} | \theta) = p(\mathcal{D} | \theta+\delta) $ iff $\delta=0$, implies that Eq 24 holds. In other words, $- E_{p(\mathcal{D})}[ \nabla_\theta^2 \log p(\mathcal{D|\theta})] $ must be **positive-definite**.  This is a strong assumption for NN problems.

There are two types of Fisher estimators when it comes to approximating the expectation via (data) samples.
1. $-E_{p(\mathcal{D})}[ \nabla_\theta^2 \log p(\mathcal{D}|\theta)]$  (As shown in Eq 2, this does not guarantee to be positive semi-definite. This work exploits this result by making this assumption in Thm 1:  $p(\mathcal{D} | \theta) = p(\mathcal{D} | \theta+\delta) $ iff $\delta=0$.  )
2. $E_{p(\mathcal{D})}[ \nabla_\theta  \log p(\mathcal{D}|\theta)  \nabla_\theta^T  \log p(\mathcal{D}|\theta)   ]$  (As shown in Eq 1, this is  always positive semi-definite. This result has been used in many existing methods such as KFAC)


---
* To avoid directly addressing the positive-definite assumption in Thm 1, the authors instead introduce a  positive-definite surrogate matrix Q.
This trick turns the one-step NGD into a two-step update (Eq 16-17).
In other words, the iteration cost can be higher compared to KFAC.
The authors should discuss the additional cost of this extra step.
Figure 2 should be plotted in terms of the wall clock time instead of the number of epochs.
Since the authors use a Kronecker structure in Q, a proper baseline in all experiments should be KFAC. Please report the wall clock time instead of the number of epochs.
The additional cost could be high if a very deep NN is used.

---
* Why is the one-step Adam update in Eq 16  good enough to approximate Eq 11? In optimization, we usually have to use a double-loop update (e.g., taking a few Adam steps) since the one-step Adam update is in general not good enough. Does taking a few more Adam steps in Eq 16 improve the performance?   The authors should discuss this point.



**Summary Of The Paper:**

In this work, the authors propose an approximated natural-gradient descent for NN via the Legendre transformation.
The authors show that the proposed methods performs similarly as K-FAC and its variants  in some medium scale problems (FACES and MNIST).

**Summary Of The Review:**

The theoretical results are weak.
* There is an implicit and key assumption in Theorem 1.
* Theorem 2 and Theorem 3 are weak/standard due to the direct assumptions on Q, which bypasses the convergence analysis of Eq 16.


Without providing additional results, in the current form, the empirical results are also weak.
I wonder if the one-step Adam update shown in Eq 16 is enough for more challenging problems.
The meta-learning approach is very similar to the auto-encoder task considered in the paper. In this case, the one-step Adam update may be good enough.
My question is whether the proposed method such as the one-step Adam update works well on other tasks such as CNN.

---

> ### Author Response · Authors · 2022-11-16
> **Concerning the assumptions of Theorem 1**
>
> We thank the reviewer for their time, and specially for lending a critical eye on our Theorem 1.
>
> Concerning the Reviewer's comment on the assumption of Theorem 1: we agree that for $p(\mathcal{D}|\theta) = p(\mathcal{D}|\theta+\delta)$ to hold true if _and only if_ $\delta=0$, the FIM must be $\succ 0$. In general, only positive semi-definiteness can be guaranteed. However, here we systematically work with the damped FIM, $\mathcal{I}(\boldsymbol\theta) + \gamma I$, in our experiments ─ as do all practitioners of second-order NN optimization, as far as we know. This damped FIM is always positive definite, such that the assumption made in Theorem 1 is in fact unnecessary. We are grateful to the Reviewer for bringing this up though, and we  decided to take this opportunity to introduce damping up front in our theoretical results. We have therefore rewritten Theorem 1, which now involves the _regularized_ cross entropy (see new Eq. 10), whose unique minimum is now guaranteed to be at $\boldsymbol\delta=0$ without further assumptions. The theory now links the Hessian of the Legendre-Fenchel transform of this regularized cross-entropy to the inverse of the damped FIM, $(\mathcal{I}(\boldsymbol\theta) + \gamma I)^{-1}$ (see revised Eq. 12).
>
> Concerning the Reviewer's remark that there are two estimators of the FIM, and that ─ on any finite data sample ─ the one we use (based on second-order derivatives) isn't necessarily positive semi-definite: although we agree with this fact, we are never using this estimator in a place where finite-sample positive semi-definiteness matters critically. We are using $Q \succ 0$ to approximate the inverse of the *average* of this estimator across minibatches, and since it is $Q$ that is used for preconditioning the gradient, it is the positive-definiteness of $Q$ that matters. Moreover, if we were running many inner-loop iterations of Eq. 16 for fixed $\boldsymbol\theta$ (which we are not -- see below our response to another of the Reviewer's comments), we would always draw a new minibatch from the model distribution in each iteration, in order to get stochastic estimates of the gradient of the average. We agree that this is where one needs to be careful not to “specialize” those inner loop updates to a single minibatch for which the FIM estimator might not be positive semi-definite! We hope this answer alleviates the Reviewer's concern?

---

> ### Author Response · Authors · 2022-11-16
> **Concerning Theorems 2 and 3 (proof of convergence)**
>
> Concerning Theorem 2 and 3 being “weak/standard” because we make “direct assumptions about the matrix $Q$”, we are not entirely sure that we correctly interpreted the Reviewer's remark. Anyhow, we would like to clarify the following points, which the Reviewer can build on if they have time to interact with us further:
>
> - First, these Theorems are not meant to provide tight estimates of the convergence rate ─ rather, they are meant to guarantee convergence (whatever its speed) under certain assumptions. We agree that stronger results (i.e. a tighter bound on $\mathcal{L}(\boldsymbol\theta_t) - \mathcal{L}(\boldsymbol\theta^*)$) could perhaps be obtained by considering the specific process by which the preconditioning matrix $Q$ evolves throughout training (i.e. by considering the specific effect of Equation 16). This is more difficult, and we do not have such a result. Is this what the Reviewer meant? We thought that proving convergence at _some_ rate would be important given that FishLeg is an entirely new method, but that a more detailed analysis of its convergence rate could perhaps be left for future work?
>
> - Second, concerning the direct assumptions we make about $Q$, FishLeg _constrains_ $Q$ to remain positive definite at all times, so this “assumption” of positive definiteness is in fact built in our definition of the  algorithm. The second assumption we make (which admittedly was not made very explicit in our submission ─ this is corrected now) is that the eigenvalues of $Q$ are bounded _uniformly across time_ by $\lambda_\min > 0$ and some $\lambda_\max$; thus, we assume that these eigenvalues do not go to zero (or else the Theorem would become meaningless) and that they do not diverge. This is indeed a genuine assumption that our $\succ 0$ parameterization of $Q$ does not explicitly enforce. In practice though, since $Q$ learns to approximate the inverse of the _damped_ FIM (c.f. above), its eigenvalues are upper-bounded by 1/gamma. As for the lower bound, one could also explicitly enforce $\lambda_\min >0$ by adding a small multiple of the identity to our $\succ 0$ parameterization of $Q$. We wonder if the Reviewer would like us to propose this explicitly?

---

> > ### Comment · Reviewer_QPAM · 2022-11-26
> > **Thanks for the response**
> >
> > I will increase my score to 6 since the authors do address and admit some key issues.
> >
> > The following points should be noted in the final version of this paper.
> >
> > * [0] I encourage the authors to release the implementation if this submission is accepted.
> >
> > * [1] The assumptions in Theorems 2 and 3 (proof of convergence) are strong. In fact, if the lowest and highest eigenvalues of a pre-conditioner (e.g. FIM or Hessian) can be globally/uniformly bounded, we can show any preconditioned gradient descent is convergent, where the convergence rate is determined by these two eigenvalues. The main challenge is how to globally/uniformly bound these eigenvalues of a pre-conditioner. In other words, the impact of Eq 16 is unknown due to the strong assumptions about the lowest and highest eigenvalues made in Thm 2 & 3. (The gradient Lipschitz and the PL (or strongly convexity) together bound the lowest and highest eigenvalues of the pre-conditioner.)
> >
> > * [2] Given that the theoretical results about the update of Eq 16 are unknown, more empirical studies including testing on larger data sets such as CIFAR 10/100 and other tasks such as CNN may be needed.
> >
> > * [3]  Given that the authors want to theoretically justify the proposed method (i.e., Thm 1-3), the following investigation is nice to have, especially if the authors do not plan to release the official implementation. It is theoretically interesting to see if taking more Adam steps at each iteration could lead to performance improvement regardless of the iteration cost.  Moreover, numerical issues could likely occur if the matrix Q is updated more frequently with a larger step size  $\alpha$ of the Adam step.

---

> ### Author Response · Authors · 2022-11-16
> **Why is the one-step Adam update in Eq 16 good enough?**
>
> It is hard to predict theoretically how many steps of Eq 16 ought to be performed in each iteration ─ this may even be problem-dependent. Our answer to this can only be empirical: we found that a single iteration was good enough for the benchmarks that we applied FishLeg to in this paper. In fact, as we mention on page 7, we found that we could even get away with updating $\boldsymbol\lambda$ every 10 iterations, with no noticeable loss of performance. Note that a similar effect was reported in previous KFAC papers, where it was shown that it is often unnecessary to update the preconditioners (i.e. to perform the costly inversion of the input and adjoint covariances) in every iteration; e.g. Goldfarb et al performed these inversions every 20 iterations only.

---

> ### Author Response · Authors · 2022-11-16
> **Per-iteration cost relative to KFAC**
>
>
> > To avoid directly addressing the positive-definite assumption in Thm 1, the authors instead introduce a positive-definite surrogate matrix Q. This trick turns the one-step NGD into a two-step update (Eq 16-17). In other words, the iteration cost can be higher compared to KFAC.
>
> We have added a supplementary note on algorithmic complexity in the Appendix (Appendix A.7). Assuming FishLeg uses the same block-diagonal Kronecker structure for $Q$ as KFAC's FIM approximation, the actual update step in FishLeg is cheaper than KFAC's, because it does not involve matrix inversions at each layer. Moreover, the big-O complexity of the FishLeg inner update of $\boldsymbol\lambda$ in this case is identical to that of the main outer step. For a densely connected network of $L$ layers with $N$ neurons each, processing data in batches of size $K$, FishLeg's complexity is $\mathcal{O}(LN^2 \min(K,N))$; in contrast, KFAC's complexity is $\mathcal{O}(LN^2 \max(K, N))$.
>
> > Since the authors use a Kronecker structure in Q, a proper baseline in all experiments should be KFAC.
>
> We spent some time during the rebuttal week implementing KFAC in the same codebase that we had used to compare FishLeg, SGDm and Adam in Figure 3 on wallclock time. We follow Goldfarb et al (2020) and perform KFAC's preconditioner inversions every 20 iterations only. Similarly, we perform the FishLeg inner-update of Eq 16 every 10 iterations only (as we were already doing). In this setup, the average per-iteration wallclock time for FishLeg is 40% (resp. 13%) smaller than for KFAC on the MNIST (resp. FACES) dataset. On the other hand, KFAC's FIM approximation is good from the start, so it makes more rapid per-iteration progress on the training loss than FishLeg for the first few tens of iterations. Overall, FishLeg and KFAC perform roughly similarly in terms of training loss vs. wallclock time. This is now in a revised Figure 3.

---

> ### Author Response · Authors · 2022-11-17
> **Note**
>
> The responses below should be read from bottom to top!

---

### Official Review · Reviewer_q97H · 2022-10-24

**Confidence:** 2
**Correctness:** 4
**Technical Novelty And Significance:** 3
**Empirical Novelty And Significance:** 3
**Recommendation:** 8

**Clarity, Quality, Novelty And Reproducibility:**

Paper is clearly written and relevant references are provided. It also provides a good account for related work, generally understandable by someone with general knowledge of optimization methods. Additional details are provided in the appendix and the codebase is made available on github with instructions to reproduce the results.

**Details Of Ethics Concerns:**

Authors consider an optimization method inspired by natural gradient descent which is based on meta-learning an inverse of the Fisher information matrix that can be optimized online during training. The paper reads well and is understandable even for someone outside of the optimization algorithms subfield.

To be more suitable for a larger audience I think the paper would benefit from an additional paragraph discussing how this method could fit into common use cases (i.e. large parameter regimes) and if there are model-specific scaling expectations in terms of runtime (compared to SGDm).


**Strength And Weaknesses:**

Strengths:
* Paper present empirical verifications on model and real benchmarks
* Proposed method features convergence guarantees
* Claims are clearly stated and accompanied by proofs

Weaknesses:
* Wall clock time is not compared directly with other second order approximations (KFAC, etc)
* Limited guidance provided on the choices of parameterization of the inverse fisher information matrix. (might be common knowledge in the field)


**Summary Of The Paper:**

The paper proposes a new optimization algorithm for deep neural networks that incorporates aspects of curvature information via meta-learning. In particular, authors model the inverse of the Fisher information matrix that appears in natural gradient descent via a set of additional parameters that are optimized jointly in an online manner. Authors empirically demonstrate that the proposed method outperforms first order optimization methods and is competitive with other higher order methods on an autoencoder benchmark. Authors also provide a proof of convergence for functions that satisfy the PL condition.

**Summary Of The Review:**

Authors consider an optimization method inspired by natural gradient descent which is based on meta-learning an inverse of the Fisher information matrix that can be optimized online during training. The paper reads well and is understandable even for someone outside of the optimization algorithms subfield.

To be more suitable for a larger audience I think the paper would benefit from an additional paragraph discussing how this method could fit into common use cases (i.e. large parameter regimes) and if there are model-specific scaling expectations in terms of runtime (compared to SGDm).

---

> ### Author Response · Authors · 2022-11-17
> **Thank you!**
>
> We thank the Reviewer for their enthusiastic appraisal of our paper.
>
> > Wall clock time is not compared directly with other second order approximations (KFAC, etc)
>
> As mentioned to the other Reviewers as well, we spent some time during the rebuttal week implementing KFAC in the same codebase that we had used to compare FishLeg, SGDm and Adam in Figure 3 on wallclock time. We follow Goldfarb et al (2020) and perform KFAC's preconditioner inversions every 20 iterations only. Similarly, we perform the FishLeg inner-update of Eq 16 every 10 iterations only (as we were already doing). In this setup, the average per-iteration wallclock time for FishLeg is 40% (resp. 13%) smaller than for KFAC on the MNIST (resp. FACES) dataset. On the other hand, KFAC's FIM approximation is good from the start, so it makes more rapid per-iteration progress on the training loss than FishLeg for the first few tens of iterations. Overall, FishLeg and KFAC perform roughly similarly in terms of training loss vs. wallclock time. This is now in a revised Figure 3.
>
> > Limited guidance provided on the choices of parameterization of the inverse fisher information matrix. (might be common knowledge in the field)
>
> This is a good point. Being acquainted with KFAC and sequels, we tend to forget that designing a good approximation to the FIM or its inverse is not something that a broad audience would know how to do a priori. Approximating the FIM (as KFAC does) instead of the inverse FIM (as FishLeg does) is highly non-trivial, and indeed it is significantly hindered by the requirement that this approximation be efficiently invertible. The authors of KFAC and sequels dedicated several pages of maths in each of their papers to derive specific structured forms for the Fisher, involving multiple layers of mathematical approximations and assumptions. One particularly noteworthy development was the proposal of a block structure for the Fisher $\mathcal{I}$ that _implicitly_ corresponded to a block tri-diagonal version of $\mathcal{I}^{-1}$. The derivation was highly non-trivial, but the authors got the intuition for a block tri-diagonal $\mathcal{I}^{-1}$ by considering the feedforward architecture of the network and drawing links to the type of parameter dependencies that arise in graphical models with similar feedforward architectures. This showed that even when one is going to parameterize $\mathcal{I}$, it is productive to first reason about $\mathcal{I}^{-1}$. In that case, Martens et al were able to work out an efficiently invertible explicit form of $\mathcal{I}$ that implicitly made $\mathcal{I}^{-1}$ block-tridiagonal -- but this was perhaps somewhat fortuitious. In general, there is no reason to expect that this approach can be made systematic. By working directly with the inverse FIM, FishLeg not only frees the user from having to guarantee efficient invertibility, but it also allows intuitions about parameter dependencies to be expressed naturally.

---

### Official Review · Reviewer_ACHH · 2022-10-25

**Confidence:** 3
**Correctness:** 3
**Technical Novelty And Significance:** 3
**Empirical Novelty And Significance:** 3
**Recommendation:** 6

**Clarity, Quality, Novelty And Reproducibility:**

This is a well-written paper with a clearly-presented method and results.

On the related work, predicting the solution to the convex conjugate has also been explored in the optimal transport community for computing the continuous Wasserstein-2 dual in the following papers:

+ [Three-Player Wasserstein GAN via Amortised Duality](https://researchmgt.monash.edu/ws/portalfiles/portal/291820310/291812754_oa.pdf)
+ [Optimal transport mapping via input convex neural networks](https://arxiv.org/pdf/1908.10962.pdf)
+ [Wasserstein-2 generative networks](https://arxiv.org/pdf/1909.13082.pdf)
+ [On amortizing convex conjugates for optimal transport](https://arxiv.org/pdf/2210.12153.pdf)

**Strength And Weaknesses:**

Strengths
+ Using duality to estimate the natural gradient update is an appealing idea and theorem 1 and the proof are great contributions
+ The results in Section 5.2 on the settings from Goldfarb et al. are convincingly executed and attain especially nice reliability and variability

Weaknesses
+ The comparisons on the deep linear networks benchmark in Figure 1 and on the wall-clock times in Figure 3 do not consider other natural gradient methods as baselines, only SGD with momentum and Adam
+ The top part of Figure 2 shows that FishLeg does not significantly improve upon the overall training loss/test error of other methods.
+ The approximation $Q(\lambda)$ may not be very accurate, especially at the beginning of training. Can bad approximations to it cause bad updates to be performed? It could be interesting to note that sub-optimal predictions of $Q(\lambda)$ could be improved by either running more updates to it or having an explicit fine-tuning phase to find a better conjugate.
+ $Q(\lambda)$ is the same shape as the inverse Fisher matrix and is thus an extremely high-dimensional prediction problem. The full (or Kronecker-factored) matrix is too memory intensive to compute for large models, so it seems challenging and computationally intensive to predict it for large models.
+ How strong are the assumptions of Theorem 1?

**Summary Of The Paper:**

This paper proposes an approximation to the natural gradient update by using Legendre-Fenchel duality to estimate the inverse Hessian with the convex conjugate of the entropy, which is summarized in Theorem 1. Section 4.2 goes on to parameterize a model $Q(\lambda)$ to predict the solution to the conjugate in (11). The method is experimentally evaluated on deep linear networks (Section 5.1) and the auto-encoding benchmarks from Goldfarb et al., 2020.

**Summary Of The Review:**

I recommend to accept the paper as it's a reasonable and well-executed contribution. I would be willing to raise my score after a discussion on not having comparisons to other natural gradient methods in some of the experiments.

---

> ### Author Response · Authors · 2022-11-16
> **Thank you and response to first couple of points**
>
> We thank the Reviewer for their time and their benevolent evaluation of our paper. Thanks a lot as well for pointing us to all these references on amortized duality in optimal transport, which we were not aware of! We are still diggesting them and thinking about how to best integrate them in our Background section ─ which we will do.
>
> > The comparisons on the deep linear networks benchmark in Figure 1 and on the wall-clock times in Figure 3 do not consider other natural gradient methods as baselines, only SGD with momentum and Adam
>
> We spent some time during the rebuttal week implementing KFAC in the same codebase that we had used to compare FishLeg, SGDm and Adam in Figure 3 on wallclock time. We follow Goldfarb et al (2020) and perform KFAC's preconditioner inversions every 20 iterations only. Similarly, we perform the FishLeg inner-update of Eq 16 every 10 iterations only (as we were already doing). In this setup, the average per-iteration wallclock time for FishLeg is 40% (resp. 13%) smaller than for KFAC on the MNIST (resp. FACES) dataset. On the other hand, KFAC's FIM approximation is good from the start, so it makes more rapid per-iteration progress on the training loss than FishLeg for the first few tens of iterations. Overall, FishLeg and KFAC perform roughly similarly in terms of training loss vs. wallclock time. This is now in a revised Figure 3.
>
> We haven't had the chance to add KFAC to Figure 1 yet but will do for the camera ready version. We expect it to converge a little faster than FishLeg (because in deep linear networks, KFAC = exact natural gradient in the limit of large batch sizes). It will indeed be interesting to have KFAC as a reference here, to better appreciate how long FishLeg takes to approximate the inverse FIM.
>
> > The top part of Figure 2 shows that FishLeg does not significantly improve upon the overall training loss/test error of other methods.
>
> Indeed, KFAC is remarkably hard to beat -- e.g. Goldfarb et al's KBFGS was also very much on par with KFAC (this may be partly due to the benchmarks that the field seems to have settled in). As we mention in the paper, we believe a major advantage of FishLeg is the greater flexibility it affords for approximating the (inverse) FIM. One needs not worry about efficient invertibility of $Q$ -- only fast $Qv$ products. FishLeg is conceptually easy to implement, and should be more amenable to modularization than KFAC or KBFGS, as it only requires the user to specify a particular parametric form for the block of Q corresponding to each neural network layer -- the rest is generic. It is early days for FishLeg but we look forward to applications to other architectures such as transformers or recurrent networks.
>
> > The approximation  may not be very accurate, especially at the beginning of training. Can bad approximations to it cause bad updates to be performed? It could be interesting to note that sub-optimal predictions of  could be improved by either running more updates to it or having an explicit fine-tuning phase to find a better conjugate.
>
> On page 6, we have a paragraph on "Initialization" where we explain that it is straightforward to initialize FishLeg's $\boldsymbol\lambda$ parameters (which determine the initial approximation to the inverse FIM) in such a way that FishLeg updates are initially identical to SGD updates (with momentum). We find that this typically gets FishLeg off to a good start. As for whether running more inner updates of $\boldsymbol\lambda$ would be beneficial: it is an interesting question! Our experiments suggest otherwise (in fact, as mentioned on page 9, we can even get away with updating $\boldsymbol\lambda$ every 10 iterations only, with no noticeable loss of performance). In Appendix A.8 (previously A.7), we also draw a link between FishLeg and a form of amortization of the conjugate gradient steps in Hessian-free optimization. In the HF literature, people have noted that it is in fact typically better *not* to run CG till convergence -- partial CG seems to act as a form of regularization. This might explain why partial minimization of our auxiliary loss works well in practice.

---

> ### Author Response · Authors · 2022-11-16
> **Response to the other points**
>
> > $Q(\lambda)$ is the same shape as the inverse Fisher matrix and is thus an extremely high-dimensional prediction problem. The full (or Kronecker-factored) matrix is too memory intensive to compute for large models, so it seems challenging and computationally intensive to predict it for large models.
>
> In block-diagonal Kronecker-factored form, $Q(\lambda)$ actually has roughly the same number of parameters as the model itself (same scaling with depth and layer size). For very large models that near-completely fill the system's memory, this is indeed impractical (but then Adam, or any method that introduces momentum, could also be ruled out on that basis). Perhaps the reviewer is more worried about the first part of their comment, namely that learning the FIM is a high-dimensional prediction problem such that it is unclear how much data is needed to do this well. This problem is partly mitigated by the block Kronecker structure that reduces the number of free parameters, but we also note (cf end of page 4) that we only need to learn the inverse FIM in the relevant subspace occupied by consecutive gradients, as opposed to learning the entirety of the FIM.
>
> > How strong are the assumptions of Theorem 1?
>
> Thank you for asking this ─ this was also asked by Reviewer QPAM, and we would like to refer you to our post below titled "Concerning the assumptions of Theorem 1". In short, when damping is introduced to the FIM, one of the assumption of Theorem 1 can be dropped (as it always hold then). The other assumption, namely that the model's log likelihood is twice differentiable, is fairly weak.

---

> ### Author Response · Authors · 2022-11-17
> **Note**
>
> The responses below should be read from bottom to top!

---

> ### Author Response · Authors · 2022-12-01
> **Thank you and we wonder if we addressed the Reviewer's concerns?**
>
> We thank again the Reviewer for the detailed evaluation of our work. We wonder if our responses and additional results addressed the Reviewer's concerns?

---

### Decision · Program_Chairs · 2023-01-20

**Decision:**

Accept: notable-top-25%

**Justification For Why Not Higher Score:**

Since the selling point is efficiency/scalability, it might have been more of a slam dunk with experiments on a large scale problem.

**Justification For Why Not Lower Score:**

See above.

**Metareview: Summary, Strengths And Weaknesses:**

This paper presents a new approach to second-order optimization of neural nets based on meta-learning an approximation to the inverse Fisher information matrix. They reformulate the FIM as the Hessian of a Legendre conjugate, and from this derive an update rule which (in principle) converges to the true inverse.

I believe this is a strong paper on all counts. The method is appealing and possibly opens up new efficient approximations to the FIM. The writing is clear, there are theoretical guarantees that apply to neural nets, and the experiments seem rigorous (if small-scale). Some reviewers question the reasonableness of the theoretical assumptions, but it seems hard to prove things about NN optimizers with much weaker assumptions. (Regarding the choice of datasets, CIFAR and ImageNet have tended not to be that interesting for optimization anyways, so the choice of datasets seems adequate to me.) The reviewers' main complaint in the first round was the lack of comparisons to second-order optimizers; the revision includes comparisons against K-FAC and K-BFGS (strong baselines) which show competitiveness and sometimes improvement. I have no reservations about acceptance.

It's probably worth discussing the relationship to amortized proximal optimization (Bae et al., 2022), which seems similar in spirit.

**Note From Pc:**

if the above contains the word "oral" or "spotlight" please see: "oral" presentation means -> notable-top-5% and "spotlight" means -> notable-top-25%. As stated in our emails, we are disassociating presentation type from AC recommendations